# Malaria parasite LIMP protein regulates sporozoite gliding motility and infectivity in mosquito and mammalian hosts

Jorge M Santos[1][†], Saskia Egarter[2], Vanessa Zuzarte-Luís[1], Hirdesh Kumar[2,3], Catherine A Moreau[2], Jessica Kehrer[2], Andreia Pinto[1], Mário da Costa[1], Blandine Franke-Fayard[4], Chris J Janse[4], Friedrich Frischknecht[2], Gunnar R Mair[1,2]*

[1]Instituto de Medicina Molecular, Faculdade de Medicina da Universidade de Lisboa, Edifício Egas Moniz, Av. Prof. Egas Moniz, Lisbon, Portugal; [2]Parasitology, Department of Infectious Diseases, University of Heidelberg Medical School, Heidelberg, Germany; [3]Kusuma School of Biological Sciences, Indian Institute of Technology Delhi, New Delhi, India; [4]Leiden Malaria Research Group, Department of Parasitology, Leiden University Medical Center, Leiden, The Netherlands

**\*For correspondence:**
gunnarmair@yahoo.com

**Present address:** [†]Department of Immunology and Infectious Diseases, Harvard T.H. Chan School of Public Health, Boston, MA, United States

**Competing interests:** The authors declare that no competing interests exist.

**Abstract** Gliding motility allows malaria parasites to migrate and invade tissues and cells in different hosts. It requires parasite surface proteins to provide attachment to host cells and extracellular matrices. Here, we identify the *Plasmodium* protein LIMP (the name refers to a gliding phenotype in the sporozoite arising from epitope tagging of the endogenous protein) as a key regulator for adhesion during gliding motility in the rodent malaria model *P. berghei*. Transcribed in gametocytes, LIMP is translated in the ookinete from maternal mRNA, and later in the sporozoite. The absence of LIMP reduces initial mosquito infection by 50%, impedes salivary gland invasion 10-fold, and causes a complete absence of liver invasion as mutants fail to attach to host cells. GFP tagging of LIMP caused a limping defect during movement with reduced speed and transient curvature changes of the parasite. LIMP is an essential motility and invasion factor necessary for malaria transmission.

## Introduction

Malaria parasites rely on gliding motility for migration through and invasion of diverse cells in two different host species: mosquitoes and humans. Ookinetes and sporozoites, the motile forms of the malaria parasite, are morphologically distinct but share core components of a gliding motility apparatus. The glideosome, an actin-myosin motor anchored within the inner membrane complex (IMC), is connected to the substrate via adhesins localised at the parasite surface (*Bargieri et al., 2014*; *Baum et al., 2008*; *Harding and Meissner, 2014*; *Kono et al., 2013*, *2012*). As shown in the related pathogen *Toxoplasma gondii*, adhesins are discharged upon contact with host cells from specialised secretory organelles, the micronemes (*Carruthers and Sibley, 1997*). Some malaria micronemal proteins, like the ookinete SOAP (*Dessens et al., 2003*) and WARP proteins (*Yuda et al., 2001*; *Li et al., 2004*), are secreted into the extracellular milieu, while others—CTRP for example (*Yuda et al., 1999*; *Dessens et al., 1999*)—are retained in the plasma membrane via transmembrane domains, from where they can engage host cell receptors. Translocated backwards along the cell surface during gliding, TM domain-containing proteins are continuously shed by rhomboid and subtilisin-like serine proteases (*Baum et al., 2008*; *Brossier et al., 2005*; *Baker et al., 2006*; *Ejigiri et al., 2012*) and deposited onto the substrate as gliding trails that can be detected by

electron microscopy (*Stewart and Vanderberg, 1992*) or immunofluorescence (*Ejigiri et al., 2012*; *Kappe et al., 1999*; *Kariu et al., 2006*). Many factors involved in adhesion, motility and invasion of ookinetes and sporozoites are largely expressed in a life-cycle-dependent manner: they include CTRP (*Yuda et al., 1999*; *Dessens et al., 1999*), WARP (*Yuda et al., 2001*; *Li et al., 2004*; *Ecker et al., 2008*), SOAP (*Dessens et al., 2003*), CHT1 (*Li et al., 2004*; *Dessens et al., 2001*) and MAOP (*Kadota et al., 2004*) in the ookinete; in the sporozoite AMA1 (*Giovannini et al., 2011*; *Bargieri et al., 2013*; *Silvie et al., 2004*), TRAP (*Kappe et al., 1999*; *Sultan et al., 1997*; *Matuschewski et al., 2002*), MAEBL (*Kariu et al., 2002*; *Saenz et al., 2008*), SPECT (*Ishino et al., 2004*), SPECT2 (*Ishino et al., 2005a*), GEST (*Talman et al., 2011*) and P52 (*Ishino et al., 2005b*; *Annoura et al., 2014*; *Labaied et al., 2007a*; *van Dijk et al., 2005*); CelTOS is common to both (*Jimah et al., 2016*). Additional surface proteins implicated in parasite motility and host cell invasion include S6/TREP/UOS3 (*Mikolajczak et al., 2008*; *Steinbuechel and Matuschewski, 2009*; *Combe et al., 2009*), TLP (*Hellmann et al., 2011*; *Moreira et al., 2008*; *Heiss et al., 2008*; *Lacroix and Ménard, 2008*), PCRMP1 and 2 (*Thompson et al., 2007*), the rhoptry-resident proteins TRSP (*Kaiser et al., 2004*; *Labaied et al., 2007b*) and RON4 (*Giovannini et al., 2011*), the GPI-anchored circumsporozoite protein (CSP) of the sporozoite and the small solute transporter PAT (*Kehrer et al., 2016a*) (*Supplementary file 1*). If, and how these different factors interact with each other, and in turn with host cell receptors to coordinate efficient gliding motility and host cell invasion is poorly understood (*Meissner et al., 2013*). Recombinant protein and peptide competition studies have revealed binding of CSP to mosquito salivary glands (*Myung et al., 2004*; *Sidjanski et al., 1997*) through mosquito heparan sulphate (*Sinnis et al., 2007*) and a TRAP-saglin interaction to allow efficient invasion of this organ (*Ghosh et al., 2009*). In the liver, heparan sulphate proteoglycans (HSPGs) on the surface of different cells but also on the extracellular matrix have been proposed to act as main receptors for sporozoites. Again, CSP and TRAP are key players in sporozoite sequestration by interacting with the cell's HSPGs (*Prudêncio et al., 2006*; *Morahan et al., 2009*; *Frevert et al., 1993*; *Coppi et al., 2007*), but also with fetuin-A on hepatocyte membranes in the case of TRAP (*Jethwaney et al., 2005*).

Gliding motility and cell invasion mechanisms are conserved among apicomplexan parasites. While intracellular motor, IMC and pellicle components are shared between *Toxoplasma gondii* and *Plasmodium* parasites (*Harding and Meissner, 2014*), many of the *Plasmodium* adhesins (14 out of 24) lack clearly recognisable homologs in *T. gondii* (*Supplementary file 1*). Here, we identify the 110 amino acid LIMP protein as a crucial surface protein for gliding motility in the rodent malaria model *P. berghei*. LIMP (the name refers to a peculiar gliding phenotype in the sporozoite arising from epitope tagging of the endogenous protein) is a key factor for sporozoite infectivity in the mosquito and the rodent host: LIMP regulates gliding motility and salivary gland invasion; it enables sporozoites to adhere to, traverse and invade host hepatocytes. Conserved in human malaria parasites and related apicomplexans, the protein could be a target for transmission-blocking vaccines in a manner developed for CelTOS (*Espinosa et al., 2017*) or TRAP (*Rampling et al., 2016*).

## Results

### *limp* is maternally supplied to developing ookinetes

In recent genome-wide studies, we have identified more than 100 mRNAs encoding known and putative surface proteins to be under translational control during transmission of *P. berghei* gametocytes from the rodent to the mosquito host (*Mair et al., 2006*, *2010*; *Guerreiro et al., 2014*). One of these transcripts encodes the protein PBANKA_0605800 (from here on forward referred to as LIMP). LIMP is encoded by an 1194 basepair long gene in the *P. berghei* ANKA strain. The gene (*Figure 1A*) comprises six exons and five introns (this organisation is conserved throughout the genus) and encodes a protein of 110 amino acids (aa) with a 22 aa long signal peptide (*Figure 1B*) and a molecular weight of 13 kDa. *Ab initio* protein structure predictions (*Combet et al., 2000*; *Xu and Zhang, 2012*) indicate that LIMP ($I^{23}$ to $G^{110}$) consists of three beta sheets opposed to two α-helices (*Figure 1C*). LIMP is highly conserved among the various *Plasmodium* species (*Figure 1B*; www.plasmodb.org) (*Aurrecoechea et al., 2009*); similarly short proteins are present in related apicomplexan parasites, where the homology is focused on a 22 amino acid proline-rich region adjacent to the signal peptide (*Figure 1—figure supplement 1*).

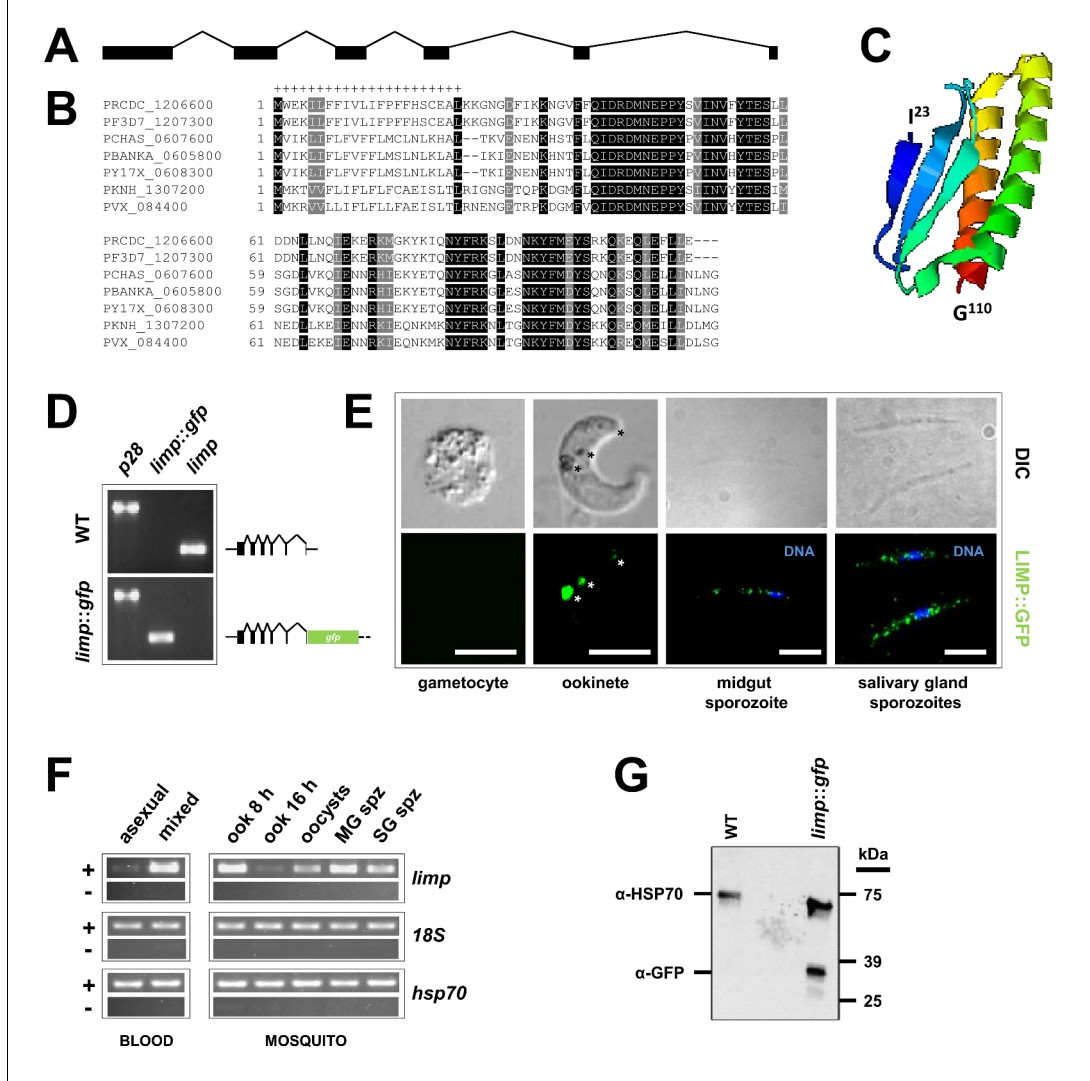

**Figure 1.** Gene and protein structure of *limp* and its translational regulation during transmission. (A) The *Plasmodium berghei limp* gene (1194 bp) is composed of six exons (shaded bars) and five introns (lines). (B) Protein alignment of LIMP orthologues from seven *Plasmodium* species. '+' signal peptide in *P. berghei*. (C) QUARK *ab initio* folding simulation model of LIMP. (D) RT-PCR analyses shows absence of WT *limp* and presence of correctly spliced *limp::gfp* mRNA in blood stages of *limp::gfp* parasites. *p28* serves as control transcript. (E) Immunofluorescence assay (IFA) of *limp::gfp* blood stages shows no LIMP::GFP expression in gametocytes. Live imaging of blood meal-retrieved ookinetes shows LIMP::GFP localisation in discrete foci (*). IFA of *limp::gfp* midgut sporozoites at day 24 p.i. and salivary gland sporozoites from day 21 p.i. shows a speckled distribution of LIMP::GFP. Parasites were stained with anti-GFP antibodies and DNA was stained with Hoechst-33342. Scale bars = 5 μm. (F) RT-PCR of *limp* through the parasite life cycle. Asexual: asexual blood stages; mixed: asexuals and gametocytes; ook: ookinetes (8 and 16 hr after gametocyte activation); MG spz: midgut sporozoites; SG spz: salivary gland sporozoites. *18S* rRNA and *hsp70* serve as loading control genes. +: RT-positive reaction; −: RT-negative reaction. (G) Western Blot analysis of parasites from 20 hr in vitro ookinete culture to verify correct C-terminal GFP tagging of LIMP. Antibodies against GFP (α-GFP) were used to visualise LIMP::GFP (37 kDa) and antibodies against parasite HSP70 (α-HSP70) served as loading control.

The following figure supplements are available for figure 1:

**Figure supplement 1.** Multiple sequence alignment of LIMP orthologues from related apicomplexan parasites.

**Figure supplement 2.** Generation and genotyping of the *limp::gfp* parasite line.

**Figure supplement 3.** *limp::gfp* parasites show no defects in liver- and blood-stage infections.

*limp* was found downregulated in a gene knock-out for the translational repressor DOZI and its mRNA enriched in an RNA-immunoprecipitation (RIP) of CITH (*Mair et al., 2006*, *2010*; *Guerreiro et al., 2014*), suggesting that *limp* mRNA is kept translationally repressed in female gametocytes until activated during ookinete development, or later. To assess experimentally when *limp* is translated, we tagged the endogenous gene at the C terminus with GFP, thus leaving the fusion under the transcriptional control of the native promoter in this haploid protozoan (*Figure 1—figure supplement 2*); then, we followed its expression. In this mutant, *limp::gfp* is the only source for LIMP (*Figure 1D* and *Figure 1—figure supplement 2*). We found no protein expression in asexual stage parasites or gametocytes; *limp::gfp* is translated in the ookinete stage and its expression is visible in crystalloids (transient and putative storage organelles of the ookinete) (*Dessens et al., 2011*) and the surface; the protein produced a faint 'dusting' on the surface of midgut and salivary gland sporozoites (*Figure 1E*; see also Figure 4). In agreement with previously published results (*Otto et al., 2014*), *limp* mRNA was not detected in asexual blood stages; mRNA is however present in gametocytes (while protein is not) and in ookinetes (where protein is present) (*Figure 1F*); this is consistent with *limp* being translationally repressed and maternally provided to the developing ookinete. Western blot analysis of ookinete material confirmed expression of a GFP fusion protein of the expected size (*Figure 1G*). Parasites expressing LIMP::GFP showed no defects in liver stage development in vitro nor during transmission to the rodent host by mosquito bite (*Figure 1—figure supplement 3*).

## *limp* gene deletion mutants suffer cumulative population loss during mosquito and rodent passage

In order to appreciate the function of LIMP for mosquito stage development and/or invasion of the *Anopheles* vector, we generated two independent gene knock-out (KO) mutant lines—Δ*limp*-a (Δa) and Δ*limp*-b (Δb)—replacing the entire open reading frame (ORF) with a pyrimethamine resistance cassette in the reference parasite line Fluo-frmg (the Fluo-frmg clone produces GFP+ male and RFP + female gametocytes) (*Mair et al., 2010*) (*Figure 2—figure supplement 1*). The absence of LIMP did not affect asexual blood-stage development, gametocyte numbers nor female to male ratios (*Figure 2—figure supplement 2*), and the capacity of Δ*limp* parasites to produce ookinetes was comparable to that of wildtype (WT) lines; in vitro zygote to ookinete conversion rates ranged from 55% to 91% (*Figure 2A*). However, following a mosquito blood meal, infection rates (oocyst numbers per mosquito) were reduced below 50% (*Figure 2B*), while the number of oocyst-derived midgut sporozoites per oocyst remained unchanged (*Figure 2C*); ookinete development and oocyst sporulation therefore do not require LIMP. Δ*limp* sporozoites did not significantly accumulate in the haemolymph (*Figure 2D*), but showed a 10-fold reduction in salivary gland invasion (*Figure 2E*). Our results demonstrate a role for LIMP during mosquito midgut colonisation by the ookinete, and a crucial function for the protein during salivary gland invasion, but not for development; neither ookinete nor sporozoite formation were affected by the absence of LIMP. We next tested whether the salivary gland invasion defect extended to invasion of the mammalian host. To this end, naive mice were infected by intravenous (i.v.) injection of 3,500 hand-dissected sporozoites (*Figure 2F*). WT-infected mice established normal blood-stage parasitaemias, while KO-infected mice never did. Mice injected i.v. with Δ*limp* salivary gland sporozoites showed no parasite load in the liver at 44 hr post-infection (p.i.) when determined by qPCR of the *P. berghei* ANKA *18S* gene (*Figure 2G*), corroborating the notion that mutants had failed to invade the liver. In vitro hepatocyte infection assays supported the in vivo data; Δ*limp* salivary gland sporozoites did not establish exoerythrocytic forms (EEFs) inside Huh7 hepatoma cells (*Figure 2H*).

## Δ*limp* sporozoites are impaired in hepatocyte transmigration, adhesion and invasion and do not glide

Our parasite transmission data clearly established a role for LIMP during the traversal of host barriers, but no role in developmental progression. Mosquito midgut colonisation was strongly affected with a 50% reduction in oocyst numbers, and infections of the mosquito's salivary glands and the rodent host's liver were almost abolished in the absence of LIMP. Therefore, the protein's role lies likely in motility. Especially, the near-absence of salivary gland sporozoites in Δ*limp* mutants suggested that LIMP is a novel and key gliding motility factor in the sporozoite. *Plasmodium* and related

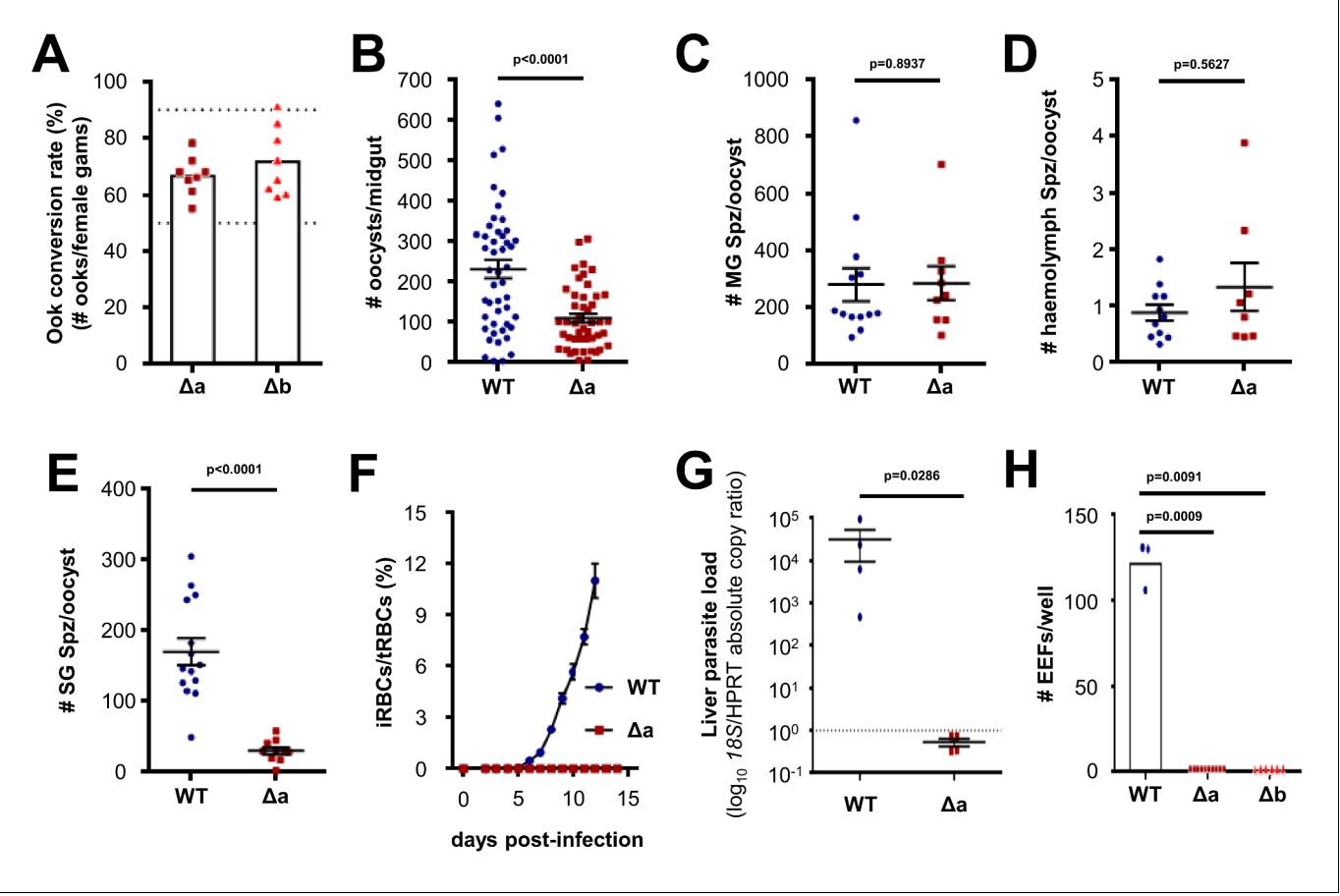

**Figure 2.** *Δlimp* parasites suffer cumulative population loss during mosquito and rodent passage. (A) Ookinete (Ook) conversion rates of both *Δ*a and *Δ*b parasites (50–90% dashed lines). Bars show means, dots represent individual conversion rates. *Δ*a and *Δ*b (four independent experiments, *n* = 8). (B) Oocysts per mosquito midgut on day 12 p.i. WT (four independent experiments; *n* = 48); *Δ*a (six independent experiments; *n* = 46). (C) Midgut sporozoites (MG Spz) per oocyst. WT (three independent experiments; *n* = 13); *Δ*a (five independent experiments; *n* = 9). (D) Haemolymph sporozoites (Spz) per oocyst. WT (three independent experiments; *n* = 11); *Δ*a (four independent experiments; *n* = 8). (E) Salivary gland sporozoites (SG Spz) per oocyst. WT (four independent experiments; *n* = 14); *Δ*a (six independent experiments; *n* = 10). (B–E) Lines show means±SEM; p-values for Mann-Whitney test. (F) Blood-stage infection following i.v. injection of sporozoites. Lines show means ± SEM. WT and *Δ*a (three independent experiments; *n* = 4). iRBCs: infected red blood cells; tRBCs: total red blood cells. (G) Parasite liver load as measured by qPCR at 44 h.p.i. Mean log10 18S/HPRT absolute copy ratio (two technical replicates) for each mouse is represented by the dots. Lines show means±SEM; p-value for Mann-Whitney test. Note that the number of 18S copies per copy of HPRT in all *Δ*a-infected mice was estimated as being below 1 (dashed line), meaning that *Δ*a parasites do not establish a successful liver infection. (H) Exoerythrocytic form (EEF) numbers during in vitro infection of hepatocytes. Bars show means; dots show individual data points; p-values for Mann-Whitney test. WT (one experiment; *n* = 3); *Δ*a (four independent experiments; *n* = 10); *Δ*b (two independent experiments; *n* = 6).

The following figure supplements are available for figure 2:

**Figure supplement 1.** Generation and genotyping of *Δlimp* parasite lines.

**Figure supplement 2.** *Δlimp* parasites develop normal blood-stage parasitaemia and gametocytaemias over the course of infection.

apicomplexan parasites move by gliding motility, which requires the establishment and dissolution of parasite-host cell contact points and the ordered turn-over of such adhesion sites (*Münter et al., 2009*); in the case of TRAP, this is mediated by rhomboid protease cleavage within the transmembrane domain severing the connection between parasite and host (*Ejigiri et al., 2012*). We therefore examined the parasite's motility behaviour in different assays: the sporozoite's ability to traverse,

invade and adhere to host hepatocytes; the quantification of CSP trails as a measure of sporozoite gliding ability; and determination of sporozoite movement patterns and speed by live microscopy.

*limp* KO sporozoites showed a complete failure to traverse hepatocytes when determined by FACS analysis; the percentage of Dextran+ host cells (the polysaccharide enters through cell membrane injury sites caused by invading sporozoites) was not different from levels obtained from cells incubated with non-infected mosquito salivary gland material (*Figure 3A*). Mutant sporozoites also showed a clear defect in adherence to Huh7 cells (*Figure 3B*) together with greatly reduced hepatocyte invasion (*Figure 3C*), explaining the failure to establish an infection in the rodent host. Next, we performed a gliding motility assay that is based on the detection of trails of the surface protein CSP, which is deposited onto microscope coverslips by moving sporozoites. KO parasites were severely impaired in gliding motility, evident by the large number of completely immotile parasites as well as the number of gliding trails of the few moving ones (*Figure 3D–E*).

Morphologically, KO parasites are normal. Transmission electron microscopy (TEM) studies of *Δlimp* sporozoites revealed no ultrastructural defects in key components of the gliding motility machinery. The arrangement of the plasma membrane (PM), inner membrane complex (IMC), sub-pellicular microtubules as well as secretory vesicles (micronemes and rhoptries) was unchanged (*Figure 3F*). As expected, *Δlimp* sporozoites were found within the mosquito's salivary gland parenchyma, confirming that some mutants can invade the salivary glands and are not merely attached to the outer surface of the glands facing the haemocoel (*Figure 3G*).

## LIMP::GFP localises to the sporozoite surface and supports wild-type life cycle progression

GFP-tagging of the endogenous gene exposed both intracellular and surface localisation (*Figure 1E*). In the ookinete LIMP::GFP highlighted, apart from faint surface staining, foci that correspond to crystalloids, transient ookinete stage-specific organelles that provide protein and lipid material for sporozoite formation in the oocyst (*Guerreiro et al., 2014*; *Dessens et al., 2011*; *Lavazec et al., 2009*; *Santos et al., 2016*; *Saeed et al., 2015*; *Lin et al., 2013*; *Saeed et al., 2013*). When we performed IFAs on *limp::gfp* sporozoites recovered from salivary glands, fluorescence was found in a dusted, speckled distribution along the entire length of the parasite (*Figure 1E* and *Figure 4A*). LIMP::GFP was detectable both with and without Triton X-100 permeabilisation (*Figure 4A*), a clear indication for parasite cell surface positioning of LIMP in a manner resembling TRAP (which is secreted from micronemes) and CSP (*Figure 4—figure supplement 1*) (*Ejigiri et al., 2012*). We confirmed this observation with two additional mutant lines, varying position and size of the tags: *mcherry::limp* (*Figure 4—figure supplement 2*) and *limp::myc* (*Figure 4—figure supplement 3*). Both showed a normal range of oocysts as well as midgut and salivary gland sporozoite numbers (*Figure 4—figure supplement 1*). The staining patterns of tagged LIMP in all three transgenic lines was identical (*Figure 4A*), showing that trafficking of tagged LIMP does not alter with tag size nor position (N- versus C-terminus), and all supporting—opposed to the knock-out—infection of the mosquito host. Using an immuno-electron microscopy (immuno-EM) approach (*Figure 4B*), we corroborated LIMP::GFP localisation in three independent staining experiments with three different parasite samples. Seventy-six percent of gold particles (from a total of 281) were associated with the parasite PM, 11% outside of parasites; 13% were intracellular although not clearly associated with any specific organelle.

Given the surface localisation of LIMP, the severe attachment and gliding defects of *Δlimp* parasites (*Figure 3B,D–E*), we wanted to examine the role of LIMP in live motility assays. Owing to the scarcity of *Δlimp* salivary gland sporozoites, we performed live in vitro gliding assays using haemolymph sporozoites instead, as these were more readily available, and compared them to WT and *limp::gfp* haemolymph sporozoites. Again, exceptionally few gene deletion sporozoites showed normal circular movement while the vast majority were found floating in the medium (*Figure 4C*), unable to attach to the glass bottom of a 96-well plate.

As expected, the *limp::gfp* haemolymph sporozoite population displayed a normal distribution of gliding motility patterns (*Figure 4C*). But when we analysed the speed of productively moving WT and *limp::gfp* sporozoites we found, surprisingly, *limp::gfp* sporozoites to move significantly slower (*Figure 4D*) despite *limp::gfp* salivary gland sporozoites producing normal numbers of CSP trails (*Figure 4E–F*). The presence of high concentrations of anti-GFP antibodies (potentially targeting LIMP::GFP) nor control IgGs affected CSP shedding (*Figure 4E*) or in vitro hepatocyte invasion by

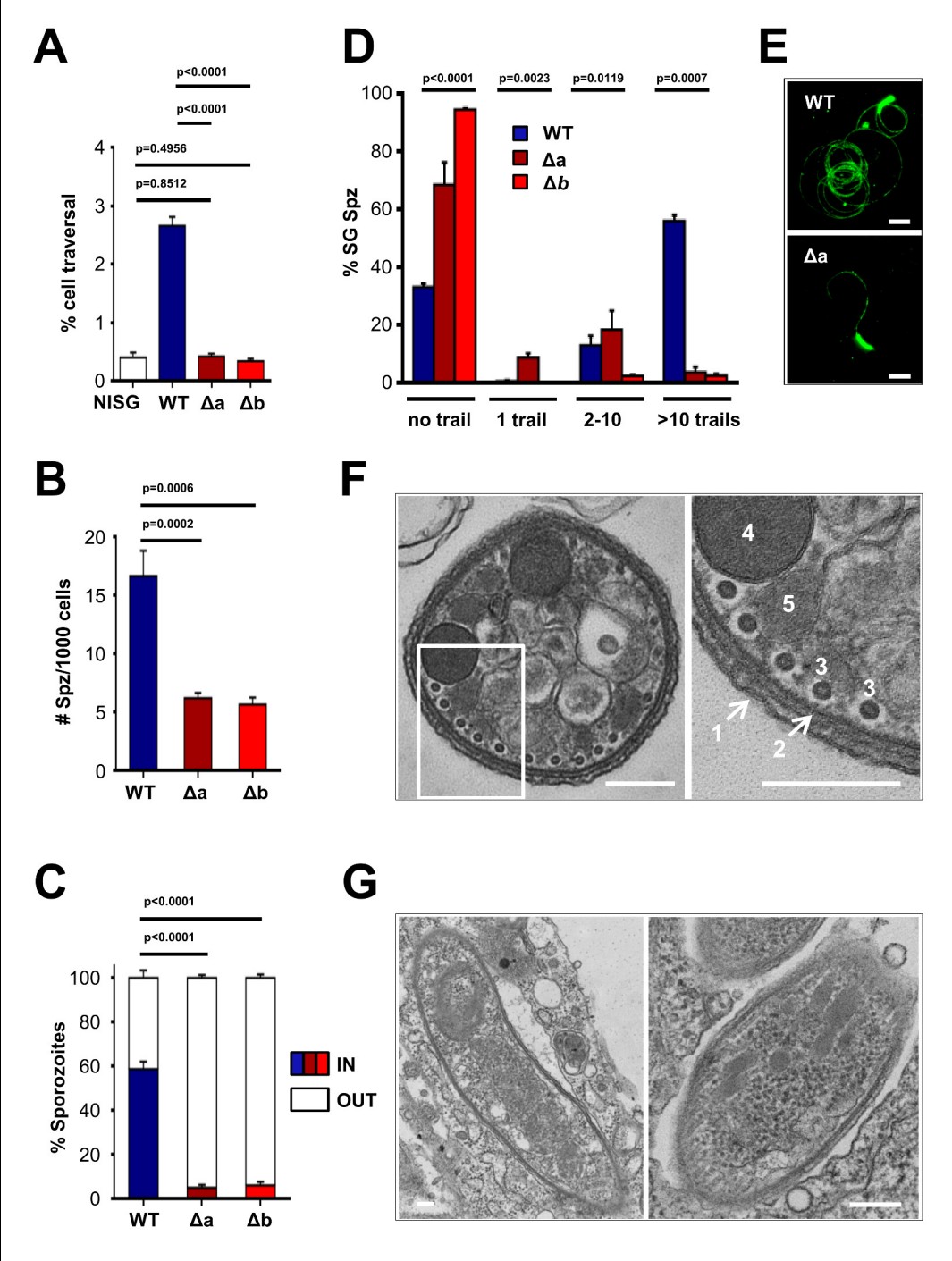

**Figure 3.** *Δlimp* parasites are severely impaired in their gliding motility, adhesion and liver cell invasion capabilities. (A) Dextran assay of sporozoite hepatocyte traversal. NISG: cells incubated with non-infected salivary gland material. WT (four independent experiments; *n* = 12); *Δ*a (five independent experiments; *n* = 10); *Δ*b (two independent experiments; *n* = 6). (B) Sporozoite (Spz) hepatocyte adhesion assay. WT (three independent experiments; *n* = 6); *Δ*a (three independent experiments; *n* = 8); *Δ*b (two independent experiments; *n* = 6). (C) Sporozoite hepatocyte invasion assay. WT (four independent experiments; *n* = 12; 1292 sporozoites analysed); *Δ*a (five independent experiments; *n* = 12; 806 sporozoites analysed); *Δ*b (two independent experiments; *n* = 6; 381 sporozoites analysed). (A–C) Bars show means±SEM; p-values for Student´s *t*-test. (D) Salivary gland sporozoite (SG Spz) gliding motility assay. Bars show means ± SEM; p-value for Kruskal-Wallis test. WT (four independent experiments; *n* = 7; 820 sporozoites analysed); *Δ*a (four independent experiments; *n* = 4; 477 sporozoites

*Figure 3 continued on next page*

*Figure 3 continued*

analysed); $\Delta$b (one experiment; *n* = 3; 299 sporozoites analysed). (**E**) Representative images of WT and $\Delta$a CSP gliding trails. Scale bars = 10 µm. (**F**) Cellular ultrastructure of $\Delta$a sporozoites is unchanged. (1) plasma membrane; (2) inner membrane complex; (3) subpellicular microtubules; (4) rhoptries; (5) micronemes. (**G**) Few $\Delta$a sporozoites are seen within the mosquito's salivary gland parenchyma, meaning they actively invaded this tissue and are thus not unviable. (**F–G**) Scale bars = 200 nm.

*limp::gfp* sporozoites (*Figure 4G*), mimicking a published TRAP study showing no effect of anti-TRAP antibodies on gliding motility (*Gantt et al., 2000*), while anti-CSP antibodies stopped sporozoite motility in vitro and blocked hepatocyte invasion in vivo (*Stewart et al., 1986*). Finally, we explored the possibility that LIMP::GFP is shed from the parasite surface during gliding in a manner resembling CSP or TRAP. To this end, we coated microscope slides with anti-GFP antibodies used for sporozoite immunofluorescence. No gliding trails were detected in this assay, suggesting that GFP-tagged LIMP is not shed from the parasite surface, or perhaps is shed at levels below the detection limit of this assay.

## LIMP::GFP parasites glide with a 'limp'

While the CSP trail-based motility analyses failed to flag the motility defect of *limp::gfp* parasites, live motility assays of salivary gland sporozoites revealed a peculiar feature in this strain: the number of moving and partially moving parasites was similar to WT (*Figure 5A*), but the speed was reduced by more than 35% when quantified by live microscopy (*Figure 5B*). Upon closer examination of trajectories of gliding paths, we identified a higher number of *limp::gfp* parasites with irregular trails and wider circumferences instead of tight, concentric rings typical for WT parasites (sporozoite tracks are visualised by connecting lines between two consecutively tracked positions) (*Figure 5C*). This means that WT sporozoites complete one full circle within a lower number of frames than the slower *limp::gfp* sporozoites. As a result, the circular tracks of WT sporozoites are tighter because the tracked positions are further apart than those of *limp::gfp* sporozoites. Typically, WT parasites move continuously in a circular fashion following their crescent shape (*Video 1*). *limp::gfp* sporozoites on the other hand produced a peculiar 'limp' (*Video 2*)—whence the gene and protein designation is derived. This limping motility is characterised by a failure of the sporozoite to detach with the rear end, while the front end continues moving forward thus leading to a transient loss of curvature and a straightening of the parasites, whereby the body size remains unchanged. *limp::gfp* sporozoites show an accentuated frequency of these cell stretching events (*Figure 5D–E*), which results in a temporary increase of sporozoite front-to-rear end distance when the parasite fails to detach (*Figure 5F*). While less than 22% of WT sporozoites exhibit one or more limps during 100 s of gliding, 62% of *limp::gfp* sporozoites limped at least three times during that period (*Figure 5D*). This effect was specific for C-terminally GFP-tagged LIMP, while both the smaller C-terminal MYC-tag or the N-terminal mCherry-tag supported WT movement and did not produce the staccato movement of *limp::gfp* (*Figure 5D–E*). Consequently, the average speed of *mcherry::limp* and *limp::myc* parasites was similar to WT and thus faster than that of *limp::gfp* sporozoites (*Figure 5B*).

## C-terminal GFP tagging occludes crucial surface areas of LIMP

The sporozoite motility data showed that C-terminal GFP tagging of LIMP resulted in a reduced detachment efficiency. LIMP contains three N-terminal beta sheets and two anti-parallel helices ($G^{60}$-$K^{83}$ and $K^{89}$-$N^{109}$) at the C-terminus (*Figure 1C* and *Figure 6—figure supplement 1*). This structure is conserved among all *Plasmodium* LIMP members (*Figure 1B*) and may provide surface and interaction sites to other parasite proteins, or proteins of the hosts. In order to compare the stability of WT LIMP structure and the effect of the different tags, we performed molecular dynamics simulations over a time period of 100 ns. The WT protein maintained a stable fold, while LIMP::GFP showed a clear destabilisation of helix 1 ($G^{60}$-$K^{83}$) resulting in a kink and a subsequent break of the helix between $H^{71}$ and $Y^{75}$ (*Figure 6—figure supplement 1*). This is exemplified by the increased distance between the C-$\alpha$ atoms of these two amino acids (*Figure 6—figure supplement 1*). Furthermore, the stable intra-helical H-bonds were lost in helix 1 ($G^{60}$-$K^{64}$, $I^{72}$-$E^{76}$ and $E^{73}$-$T^{77}$) and helix 2 ($S^{95}$-$K^{99}$) of LIMP::GFP (*Figure 6—figure supplement 1*). In addition, the GFP tag occludes the

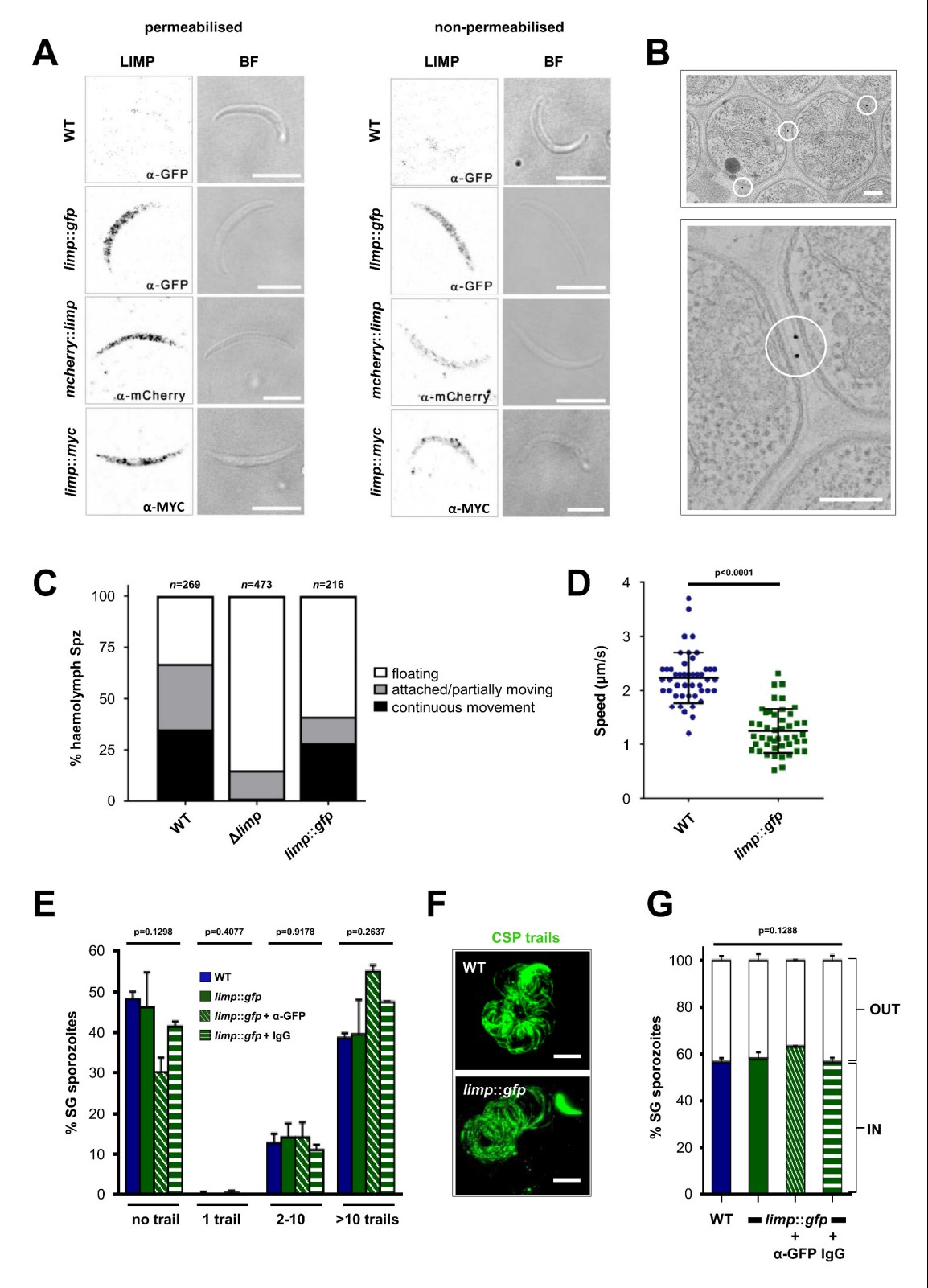

**Figure 4.** LIMP localises to the parasite plasma membrane but its function cannot be targeted by blocking antibodies. (A) Immunofluorescence assays of WT, *limp::gfp*, *mcherry::limp* and *limp::myc* salivary gland sporozoites. Sporozoites were stained with anti-GFP (α-GFP), α-mCherry or α-MYC antibodies at days 17–20 p.i. BF=brightfield images. Scale bars = 5 μm. (B) Immuno-electron microscopy of *limp::gfp* salivary gland sporozoites with anti-GFP antibodies identifies LIMP::GFP (black dots inside white circles) at the parasite plasma membrane. Scale bars = 200 nm. (C) Live motility assay

*Figure 4 continued on next page*

*Figure 4 continued*

to investigate movement patterns of WT, *Δlimp* and *limp::gfp* haemolymph sporozoites (Spz). Most *Δlimp* sporozoites display a severe attachment phenotype and thus are incapable of moving. The indicated total number of sporozoites derived from three independent experiments in case of WT and six or two independent assays were performed for *Δlimp* or *limp::gfp,* respectively. (D) Moving *limp::gfp* haemolymph sporozoites show reduced gliding speed. Dot plots show means±SEM of 46 analysed sporozoites from three independent experiments per parasite line; p-value for Mann-Whitney test. (E) CSP shedding-based gliding motility assay of WT and *limp::gfp* salivary gland (SG) sporozoites in the presence and absence of anti-GFP (α-GFP) blocking antibodies or control IgGs. Bars show means±SEM. WT (one experiment; *n* = 3; 280 sporozoites analysed); *limp::gfp* (two independent experiments; *n* = 5; 492 sporozoites analysed); *limp::gfp* + α-GFP (one experiment; *n* = 3; 197 sporozoites analysed); *limp::gfp* + IgG (one experiment; *n* = 3; 234 sporozoites analysed). (F) Representative images of WT and *limp::gfp* (no antibody) CSP gliding trails. Scale bars = 10 µm. (G) WT and *limp::gfp* salivary gland (SG) sporozoite hepatocyte invasion assay in the presence and absence of anti-GFP (α-GFP) blocking antibodies or control IgGs. Bars show means±SEM of invading (IN) and non-invading (OUT) parasites. Data are from one experiment (*n* = 3) for all parasite lines and conditions. Numbers of sporozoites analysed: WT = 346, *limp::gfp* = 763, *limp::gfp* + α-GFP = 807, *limp::gfp* + IgG = 742. (E and G) p-values for Kruskal-Wallis test.

The following figure supplements are available for figure 4:

**Figure supplement 1.** Phenotypic characterisation of tagged *limp* parasite lines.

**Figure supplement 2.** Generation and genotyping of the *mcherry::limp* parasite line.

**Figure supplement 3.** Generation and genotyping of the *limp::myc* parasite line.

surface area of LIMP helices, while neither MYC nor mCherry tags interfere with this surface (*Figure 6* and *Figure 6—figure supplement 2*). The loss of structural integrity and solvent accessibility in the GFP fusion protein is likely the cause for the reduced efficiency in detachment and limping motility.

## Discussion

Here, we have identified a novel *Plasmodium* protein, LIMP, with a core function for gliding motility and infection in the rodent *P. berghei* malaria parasite model. While genes encoding proteins involved in invasion and gliding motility are assumed to be transcribed and translated when needed, recent studies highlighted many mRNAs encoding motility and invasion proteins to be transcribed and translationally repressed in the female gametocyte, and thus maternally provided to the developing ookinete and oocyst (*Ecker et al., 2008*; *Guerreiro et al., 2014*; *Santos et al., 2016*; *Saeed et al., 2013*; *Rao et al., 2016*; *Sebastian et al., 2012*; *Raine et al., 2007*). LIMP is one of these factors. Ookinetes that lack LIMP displayed a 50% reduction in parasite burden in mosquito midguts. The resulting haemolymph sporozoites were heavily affected in their motility and ability to invade and parasitise the mosquito salivary glands; the small population of salivary gland sporozoites was incapable of establishing a productive infection in hepatocytes, neither in vitro nor in vivo. In sporozoites, LIMP localises to the parasite plasma membrane, as determined by examination of three different tagged parasite lines (*limp::gfp*, *mcherry::limp* and *limp::myc*) and immuno-gold EM studies. The protein is probably of low abundance, and LIMP is yet to be discovered in the sporozoite proteomes from the rodent parasite species *P. berghei* and *P. yoelii* or the human parasite *P. falciparum* (*Swearingen et al., 2016*; *Hall et al., 2005*; *Lasonder et al., 2008*; *Lindner et al., 2013*).

We propose that LIMP could be involved in the attachment of sporozoites to the substrate, and perhaps regulate efficient turnover of attachment sites in gliding parasites. The underlying requirement for gliding, cell traversal and invasion of malaria parasites is adhesion to the substrate, either artificial ones such as glass or plastic, or cells from various host tissues (midgut and salivary gland epithelia of the mosquito; mammalian skin, liver sinusoidal cells or hepatocytes) (*Singer and Frischknecht, 2012*; *Vaughan et al., 2008*). In *Plasmodium* sporozoites—the key model for understanding gliding motility—adhesion is a three-step process: initial attachment at the apical (front) or posterior (rear) end followed by a secondary adhesion site at the opposite end; the apical end then moves slightly forward and the sporozoite flips over on one of its sides, thus increasing the area of contact between the sporozoite body and the substrate; finally, a tertiary attachment point is made at the centre of the parasite. The primary step of adhesion is dependent on intracellular actin dynamics, as the induction of actin polymerisation leads to weaker adhesion (*Hellmann et al., 2013*;

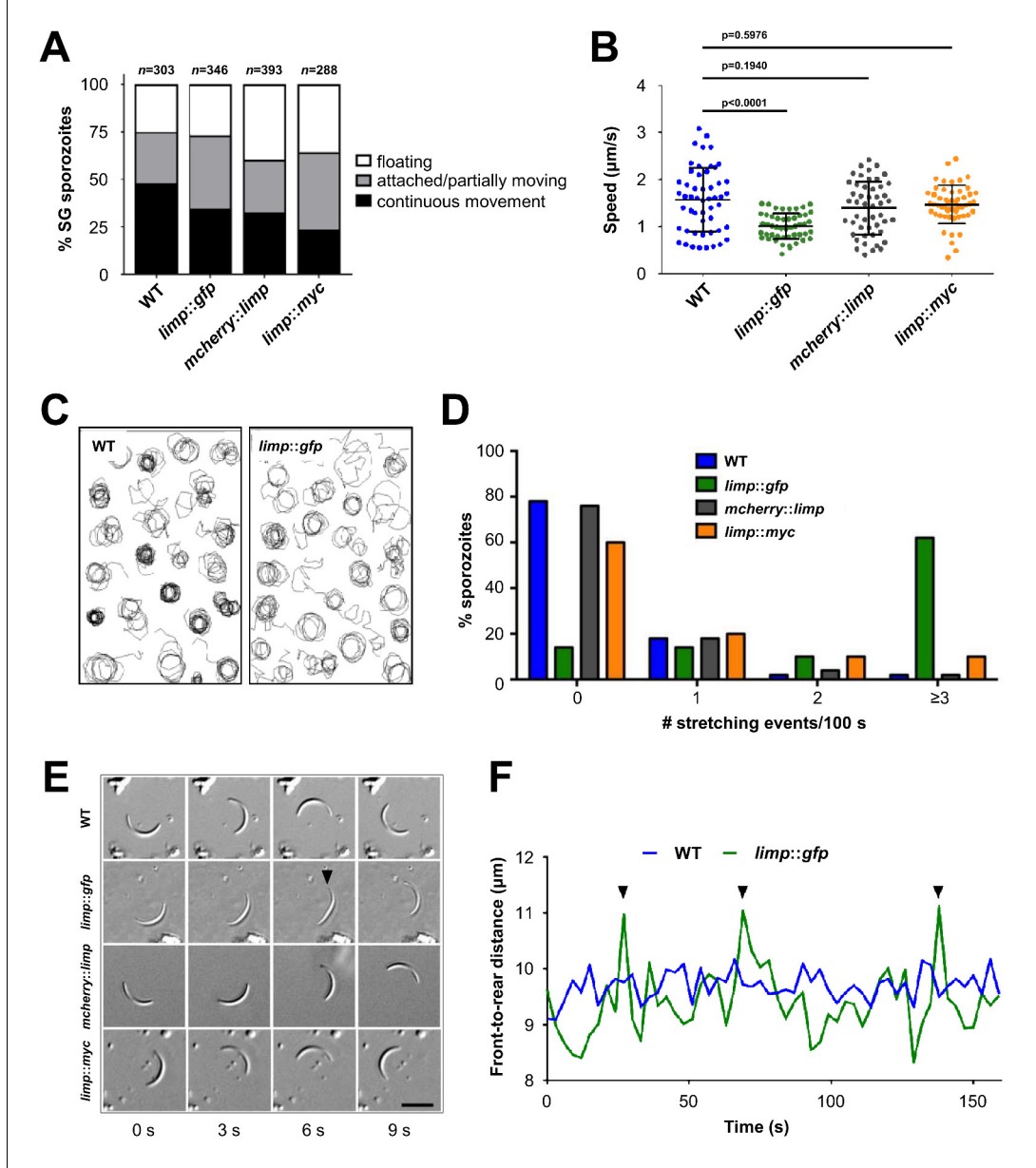

**Figure 5.** *limp::gfp* salivary gland sporozoites glide with a 'limp'. (A) The percentage of circular moving *limp::gfp* salivary gland (SG) sporozoites is comparable to WT, *mcherry::limp* and *limp::myc* sporozoites. Four independent experiments were performed for each parasite line to obtain indicated numbers of sporozoites. (B) Moving *limp::gfp* salivary gland sporozoites show significantly reduced gliding speed, whereas neither *mcherry::limp* nor *limp::myc* parasites reveal a speed reduction. Lines show means±SD; p-values for one-way ANOVA with Dunnett's post-hoc test. For each line, 50–53 sporozoites were analysed. Data for all lines represent three independent experiments. (C) Maximum projections of a representative subset of WT and *limp::gfp* moving sporozoites tracked in (B). (D) *limp::gfp* sporozoites show higher frequency of stretching events compared to WT, *mcherry::limp* and *limp::myc* parasites. Percentage of sporozoites within each stretching frequency category are shown by the bars. Fifty sporozoites were analysed per parasite line. Data for all lines are from three independent experiments. (E) Consecutive bright-field images of representative gliding WT, *limp::gfp*, *mcherry::limp* and *limp::myc* sporozoites. Arrowhead indicates the apical tip of a stretched sporozoite. Scale bar = 10 µm. (F) Distance between front and rear ends of one representative sporozoite for WT and *limp::gfp*, respectively. The distance was analysed over 159 s. Arrowheads indicate stretching events of *limp::gfp* parasites.

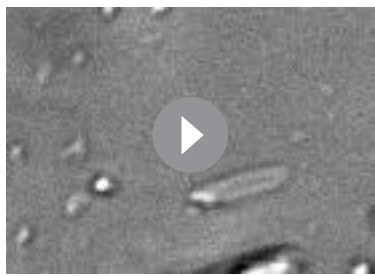

**Video 1.** Live gliding motility of a representative wild-type sporozoite

*Hegge et al., 2012*, *2010*), and requires cell surface proteins such as TRAP and S6/TREP/UOS3; they are important for initial in vitro adhesion and this function is associated with the interaction of the proteins' von-Willebrand-factor-A extracellular adhesive domains with the substrate (*Hegge et al., 2010*). TRAP-deficient parasites adhere preferentially with their front end, while sporozoites lacking S6/TREP/UOS3 show no preference for apical or rear end initial adhesion. Moreover, their differential spatial interaction with actin filaments suggests these proteins cannot compensate for one another during adhesion and motility (*Hegge et al., 2012*). Another protein, TLP, solely stabilises newly formed adhesion

sites and/or increases the speed of adhesion formation, thereby augmenting continued adhesion formation during gliding (*Hegge et al., 2010*). HSP20 has been implicated in the regulation of the dynamics of parasite adhesion site formation and rupture. Parasites lacking HSP20 show longer adhesion sites and slower gliding motility, while in vitro cell traversal and invasion is unaffected (*Montagna et al., 2012*). Δ*limp* parasites adhere less to hepatoma cells or glass surfaces implying that LIMP is required for the establishment of initial adhesion contacts with the substrate, thereby affecting the subsequent events of motility and invasion; in that sense, it has an equivalent function to TRAP (*Münter et al., 2009*).

After attachment to the substrate, continuous parasite motility requires the continuous turnover of adhesion sites. Strong adhesion forces at the rear end of the sporozoite during gliding can lead to temporal arrest, causing stretching of the parasite when it pulls forward (*Münter et al., 2009*). Unlike Δ*trap* mutants, the *limp::gfp* mutant is able to glide, albeit with reduced speed caused by frequent stretching events as a result of a de-adhesion defect at its rear end. Fusing GFP to LIMP allowed us to uncover, serendipitously, this additional function for the protein. However, the fact that *limp::gfp* parasites complete the life cycle (no differences exist in parasite numbers in mosquitoes or in prepatent period after mosquito bite) suggests that LIMP::GFP still exerts its essential role in sporozoite biology. One can therefore also be confident that its localisation at the sporozoite PM reflects the true site of action of untagged LIMP protein. Tagging of LIMP with two alternative tags of different sizes fused to either the N- or C-terminus (mCherry and MYC, respectively) replicated the subcellular localisation of LIMP::GFP, and shows that the limping phenotype is specifically induced by the large GFP tag positioned at the C-terminus of the protein; both mCherry and MYC-tagged lines behaved as WT parasites in every way analysed. We have attempted to generate anti-LIMP antibodies in rabbits to further corroborate LIMP localisation but the resulting sera failed to reveal antigen-specific, reliable staining.

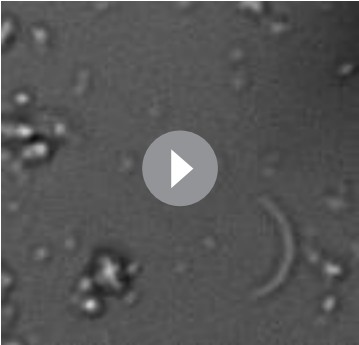

**Video 2.** Live gliding motility of a representative *limp::gfp* sporozoite

The phenotype of *limp::gfp* parasites highlights the importance of LIMP for coordinated and timely detachment of the sporozoite from its substrate, thereby allowing uninterrupted, smooth gliding. Perhaps, LIMP is a regulator of TRAP, or acts in concert with TRAP and related family members. Alternatively, LIMP could be acting entirely independently. Both scenarios highlight how little we understand about formation of adhesion sites, their turnover and the generation of force within the parasite, or how different surface proteins co-operate. LIMP is essential for parasite transmission by facilitating productive gliding motility required for the traversal of mosquito and mammalian tissues, as well as invasion of the mosquito salivary glands and liver cells in the vertebrate host (*Figure 7*).

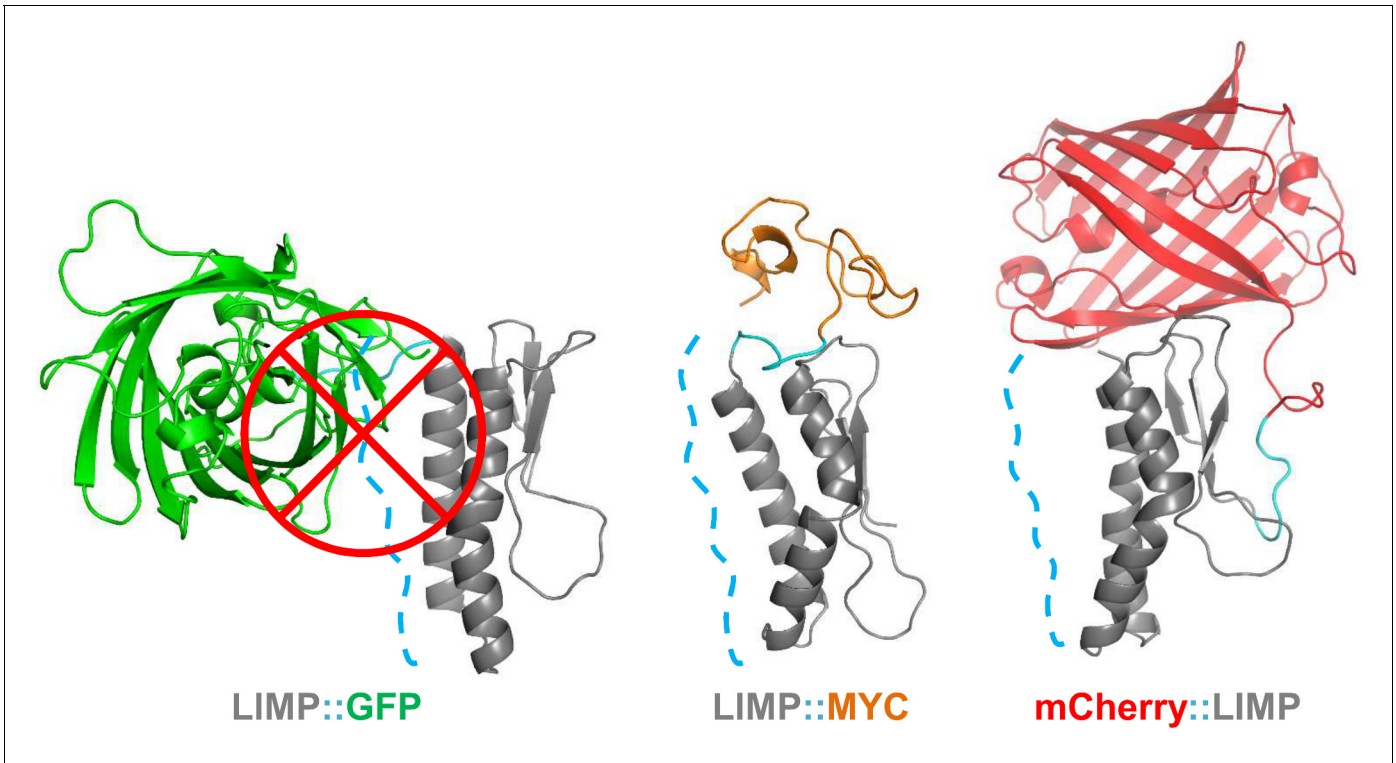

**Figure 6.** GFP tag occludes solvent accessible surface area of LIMP helices. Modelled LIMP structures linked to different tags (GFP in green, MYC in orange and mCherry in red) extracted after 100 ns of all-atom molecular dynamics simulations. Solvent accessible surface area of LIMP helices is shown as a blue dashed line. Unlike MYC or mCherry tags that do not interact with LIMP helices, GFP tag interacts with LIMP helices and thus occlude their solvent accessible surface area.

The following figure supplements are available for figure 6:

**Figure supplement 1.** LIMP secondary structure prediction and destabilisation upon GFP tagging.

**Figure supplement 2.** All-atom molecular dynamics simulations of LIMP tagged with GFP, MYC and mCherry.

Several secreted proteins of parasite origin play a role in one or more of these processes: CTRP (*Yuda et al., 1999*; *Dessens et al., 1999*; *Templeton et al., 2000*), MAOP (*Kadota et al., 2004*) and SOAP (*Dessens et al., 2003*) in the ookinete; MAEBL (*Kariu et al., 2002*; *Saenz et al., 2008*), SPECT (*Ishino et al., 2004*), SPECT2 (*Ishino et al., 2005a*), P52 (*Ishino et al., 2005b*; *Labaied et al., 2007a*) and TRAP (*Ejigiri et al., 2012*; *Kappe et al., 1999*; *Sultan et al., 1997*; *Matuschewski et al., 2002*; *Mota et al., 2001*) in the sporozoite; and CelTOS (*Kariu et al., 2006*) in both the ookinete and the sporozoite (*Supplementary file 1*). Finding LIMP binding partners will constitute an important first step to unravelling its precise molecular mechanism of action and will rely on co-immunoprecipitation or proximity-dependent biotinylation approaches (*Kehrer et al., 2016b*). LIMP is conserved across different malaria parasite species, and proteins with lower homology are present in apicomplexan parasites of humans (*T. gondii*) and livestock (*Neospora* or *Eimeria*). The protein is thus an obvious target for intervention strategies that impede the gliding motility machinery.

## Materials and methods

### Experimental

In Instituto de Medicina Molecular (iMM, Lisbon, Portugal), animal experimentation protocols were approved by the iMM Animal Ethics Committee (under authorisation

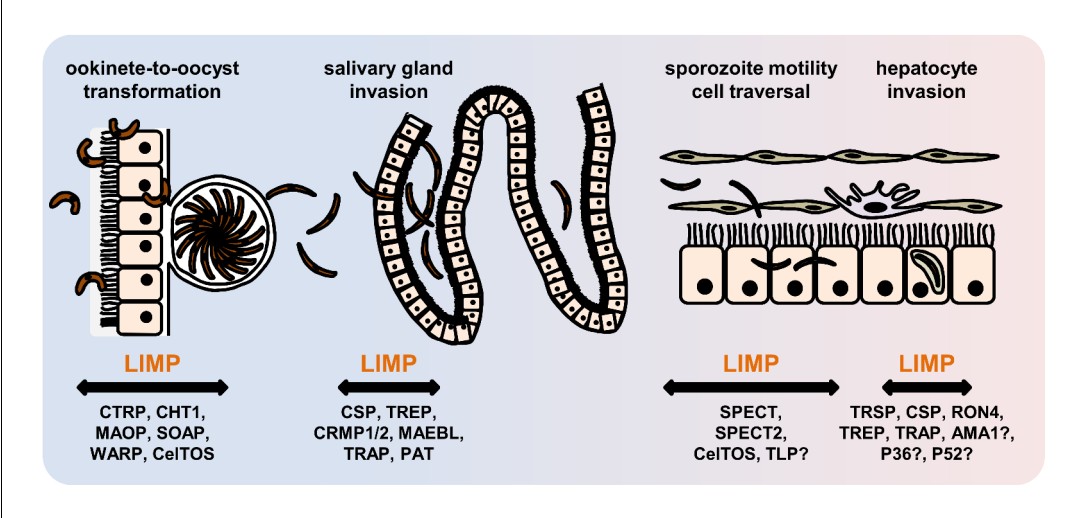

**Figure 7.** Role of motility- and invasion-related *Plasmodium* factors during mosquito stage development and transmission of malaria parasites. While previously published malaria proteins participate in just one or few motility and invasion steps of *Plasmodium* mosquito stages, LIMP plays a role in midgut invasion/oocyst formation, salivary gland invasion, sporozoite adhesion and gliding, and traversal and invasion of hepatocytes.

AEC_2010_018_GM_Rdt_General_IMM) and the Portuguese authorities (Direção Geral de Alimentação e Veterinária). Animal experiments performed in Leiden University Medical Center (LUMC, Leiden, The Netherlands) were approved by the Animal Experiments Committee of the Leiden University Medical Center (DEC 10099; 12042; 12120). At the University of Heidelberg Medical School (Heidelberg, Germany) animal work was approved by the German authorities (Regierungspräsidium Karlsruhe; 35–9185.81/G-266/16) and performed in compliance with FELASA guidelines and regulations. All animal work was in accordance with EU regulation. Animals used were female Balb/c (RRID:IMSR_CRL:28), OF-1 (RRID:IMSR_CRL:612) and NMRI (RRID:IMSR_CRL:605) mice (6–8 weeks old) bred at Charles River or Janvier Labs, France.

## Reference *P. berghei* ANKA lines

Four *P. berghei* ANKA parasite lines were used (for details see RMgm database at www.pberghei. eu): line HPE, a non-gametocyte producer clone (*Janse et al., 1989*); line 820cl1m1cl1 (Fluo-frmg; RMgm-164) (*Mair et al., 2010*) expressing RFP under the control of a female gametocyte-specific promoter and GFP under the control of a male gametocyte-specific promoter; line 259cl1 (PbGFPcon; RMgm-5) (*Franke-Fayard et al., 2004*) expressing GFP under the control of the constitutive *eef1a* promoter; and line cl15cy1, which is the reference parent line of *P. berghei* ANKA (*Janse et al., 2006a*).

## Cell lines

Huh7 cells (RRID:CVCL_0336) were obtained from the Japanese Collection of Research Bioresources Cell Bank (JCRB Cell Bank), from the National Intitutes of Biomedical Innovation, Health and Nutrition (NIBIOHN). General information about these cells can be found here: http://cellbank.nibiohn. go.jp/~cellbank/en/search_res_det.cgi?ID=385#. Huh7 is an immortal cell line derived from a well-differentiated human hepatocellular carcinoma. It has an epithelial-like morphology and was originally isolated from a liver hepatoma tissue in a 57-year-old Japanese male in 1982. The line was established by Nakabayashi and collaborators (*Nakabayashi et al., 1982*). Cell authenticity was confirmed by the Wellcome Trust Sanger Institute under the Catalogue Of Somatic Mutations In Cancer (COSMIC) Cell Lines Project. Identity data, including STR profiling, is available at: http://cancer. sanger.ac.uk/cell_lines/sample/overview?id=907071#overview.

This cell line is not listed in the International Cell Line Authentication Committee (ICLAC) Database of Cross-Contaminated or Misidentified Cell Lines. It was regularly checked throughout the

duration of this study for mycoplasma contamination using the VenorGeM OneStep – Mycoplasma Detection Kit for conventional PCR (Minerva Biolabs GmbH, #11-8025) and no evidence of contamination was ever found.

## Reverse transcriptase-PCR (RT-PCR)

To investigate the transcription pattern of *limp* by RT-PCR, RNA from different life cycle stages were obtained using TRIzol Reagent (Ambion, #15596). Reverse transcription was primed with random primers and oligo-d(T) using SuperScript II Reverse Transcriptase (Invitrogen, #18064). Oligonucleotide primers used in life cycle RT-PCRs are shown in *Supplementary file 2A*.

## Generation of *limp* gene deletion mutants

Deletion of *limp* (PBANKA_0605800) was performed with a standard replacement construct (*Janse et al., 2006b*) using a modified plasmid pL0001 (www.mr4.com), which contains the pyrimethamine-resistant *Toxoplasma gondii* (*tg*) *dhfr/ts* as a selectable marker cassette. See *Figure 2— figure supplement 1* for the name and details of the construct pLIS0060. Target sequences for homologous recombination were PCR-amplified from *P. berghei* WT genomic DNA using primers specific for the 5′ or 3′ flanking regions of *limp* (see *Supplementary file 2B* for the sequence of the different primers). The PCR–amplified target sequences were cloned in plasmid pL0001 either upstream or downstream of the resistance cassette to allow for integration of the construct into the genomic target sequence by homologous recombination after digestion with *Asp*718I and *Not*I. Transfection into Fluo-frmg parasites (*Mair et al., 2010*), selection and cloning of mutant parasite lines were performed as described (*Janse et al., 2006b*). Deletion of *limp* was confirmed by diagnostic PCR (for primers see *Supplementary file 2C*) and Southern analysis of FIGE (field inversion gel electrophoresis)-separated chromosomes (*Figure 2—figure supplement 1*) with a probe recognising the 3′ UTR of *pbdhfr/ts*. Absence of *limp* mRNA was determined by RT-PCR analysis (*Figure 2—figure supplement 1*; see *Supplementary file 2C* for primers used for RT-PCR). Two independently derived, cloned lines were used for phenotype analyses: 2091cl2m5 (Δ*limp*-a) and 2090cl2m6 (Δ*limp*-b), both in the Fluo-frmg background.

## Generation of transgenic lines expressing GFP-, mCherry- and MYC-tagged LIMP

In situ C-terminal GFP tagging of *limp* was performed by single cross-over homologous recombination (see *Figure 1—figure supplement 2* for details of the construct pLIS0079). Primers used to amplify the targeting region of *limp* are listed in *Supplementary file 2B*. The targeting region was cloned in frame with *gfp*. Linearised plasmid (*Afl*II) was transfected into cl15cy1 parasites followed by limiting dilution cloning (*Janse et al., 2006b*), resulting in the transgenic line 2180cl1m4 (*limp:: gfp*). Integration of the plasmid was confirmed by diagnostic PCR (for primers see *Supplementary file 2C*) and Southern analysis of FIGE-separated chromosomes using a probe for the 3′ UTR of *pbdhfr/ts* (*Figure 1—figure supplement 2*). Transcription of the *gfp* fusion gene was confirmed by RT-PCR (*Figure 1—figure supplement 2*). Primers used for RT-PCR are listed in *Supplementary file 2C*.

To generate the parasite clone expressing N-terminally mCherry-tagged *limp* (C505.1), the *mcherry* ORF was integrated downstream of the signal peptide of *limp*. To ensure correct cleavage of the signal peptide, the sequence after the cleavage site was repeated downstream of the *mcherry* ORF. Endogenous 5′ UTR and 3′ UTR served as targeting regions for double homologous recombination (see *Figure 4—figure supplement 2* for details of the plasmid construct pLIS0505). Integration of *mcherry::limp* into the *limp* locus of cl15cy1 parasites was confirmed via analytical PCR (*Figure 4—figure supplement 2*; for primers see *Supplementary file 2C*) of *mcherry::limp* parasites after limiting dilution cloning (*Janse et al., 2006b*).

C-terminal MYC tagging of *limp* was performed analogously to GFP tagging by single cross-over homologous recombination (See *Figure 4—figure supplement 3* for details of the plasmid construct pLIS0472). Briefly, the *limp::gfp* construct was used as starting vector and *gfp* was replaced by a triple *myc* tag. Akin to *limp::gfp*, linearised *limp::myc* plasmid was transfected into cl15cy1 parasites followed by limiting dilution cloning (*Janse et al., 2006b*) producing clone C472.1. Correct

integration of the plasmid was confirmed via PCR (*Figure 4—figure supplement 3*; for primers see *Supplementary file 2C*).

## In vitro generation and purification of ookinetes for Western blot analysis

Blood of an infected mouse was harvested by cardiac puncture and transferred to ookinete medium [250 mL RPMI 1640 supplemented with 25 mM HEPES and L-Glutamine (Gibco, #22400089); 12.5 mg hypoxanthine; 2.5 mL penicillin-streptomycin; 0.5 g NaHCO$_3$; 5.12 mg xanthurenic acid; 20% FBS] at 19°C. After 20 hr, ookinetes were purified using a 63% Nycodenz cushion. After purification of ookinetes, uninfected red blood cells were removed by lysis with 0.17 M NH$_4$Cl. The ookinete pellet was then incubated with RIPA buffer (50 mM Tris–HCl pH 7.5; 150 mM NaCl; 1% Triton X–100; 0.5% sodium deoxycholate; 0.1% SDS; 1 mM EDTA) for 1 hr on ice to lyse the parasites. Samples were then centrifuged for 30 min at 14,000 rpm at 4°C and Laemmli buffer was added to the supernatant.

## Analysis of Δ*limp* blood-stage development

Naive mice were infected intravenously (i.v.) with $10^4$ infected red blood cells (iRBCs) of either WT (Fluo-frmg) or Δ*limp* parasites. Total parasitaemia, female and male gametocytaemias were determined daily by FACS. Total parasitaemia was determined using the vital DNA dye Vybrant DyeCycle Ruby stain (Molecular Probes, #V10273). Given that both WT and Δ*limp*-a parasite lines express RFP in female gametocytes and GFP in male gametocytes, female gametocytaemias were calculated as RFP+ RBCs/total RBCs, while male gametocytaemias were calculated as GFP+ RBCs/total RBCs.

## Ookinete formation assays

Ookinete formation assays were performed following published methods using gametocyte-enriched blood collected from mice treated with phenylhydrazine/NaCl (*Beetsma et al., 1998*). The ookinete conversion rate is defined as the percentage of female gametes that develop into mature ookinetes determined by counting female gametes and mature ookinetes in Giemsa-stained blood smears 16–18 hr after in vitro induction of gamete formation.

## Oocyst and sporozoite production

Oocyst and sporozoite production of the mutant parasites was analysed by performing standard mosquito infections. Naive mice were infected intraperitoneally (IP) with $10^6$ infected red blood cells (iRBCs) of each line. On days 4–5 p.i., mice were anaesthetised and *Anopheles stephensi* female mosquitoes allowed to feed for 30 min. Twenty-four hours after feeding, mosquitoes were anaesthetised by cold shock and unfed mosquitoes were removed. Oocyst and sporozoite numbers were counted at days 11–12 and 17–21 after mosquito infection, respectively. Oocysts were counted after mercurochrome staining. Sporozoites were counted in pools of 4–10 mosquitoes.

## Transmission experiments

Transmission of Δ*limp*-a parasites was assessed by injecting i.v. 3,500 salivary gland sporozoites dissected at day 20 p.i. into naive mice. Blood-stage parasitaemias were followed up to 14 days post-injection. Livers were extracted at 44 h.p.i. and parasite load determined by quantitative real-time PCR (qPCR) using specific primers for *P. berghei* ANKA *18S* rRNA gene. Mouse hypoxanthine-guanine phosphoribosyltransferase (HPRT) was used as host tissue control gene. Primers used for qPCR are listed in *Supplementary file 2A*. Absolute number of mRNA copies was determined for each gene using standard curves.

## In vitro sporozoite gliding motility assays

WT (Fluo-frmg or cl15cy1), Δ*limp*-a, Δ*limp*-b and *limp::gfp* salivary gland sporozoites were dissected at days 20–21 p.i. and gliding motility assessed in a standard gliding assay (*Liehl et al., 2010*). Briefly, glass coverslips were coated overnight at RT with a 10 μg/mL suspension of 3D11 mouse anti-*P. berghei* circumsporozoite protein (CSP) monoclonal antibody (RRID:AB_2650479) (*Yoshida et al., 1980*). On the following day, 10,000–20,000 salivary gland sporozoites of each line were loaded per coverslip and incubated for 1 hr at 37°C, 5% CO$_2$, in the presence of 10% FBS. In

the case of *limp::gfp*, sporozoites were also pre-incubated for 1 hr on ice in the presence of rabbit polyclonal anti-GFP (Abcam, #ab6556; 10 µg/mL) (RRID:AB_305564) or control rabbit IgGs (Abcam, #ab37415; 10 µg/mL) (RRID:AB_2631996) to evaluate possible gliding blockade. Sporozoites were then fixed with 4% PFA/PBS and an anti-PbCSP immunofluorescence assay (IFA) was performed to detect CSP trails (see below for IFA details). Coverslips were analysed in a Leica DM5000B fluorescence microscope. Each sporozoite was categorised according to the number of trails it formed. Percentages of sporozoites in each category for each line were calculated as the ratio between the number of sporozoites in each category and the total number of sporozoites analysed. To perform live motility assays of haemolymph sporozoites, mosquitoes were dissected on days 15–16 p.i. To obtain haemolymph sporozoites, mosquitoes were cooled on ice for 20 min. After removing the last segment of the abdomen, the thorax was pierced with a glass pipette and the haemolymph was flushed out of the mosquito. Motility was assessed by imaging sporozoites in a Nunc MicroWell 96-well plate with optical bottom (Thermo Scientific) using an inverted Zeiss Axiovert 200M widefield microscope with an XBO75 xenon lamp, the AxioVision 4.7.2 software, a 25X objective (LCI Plan-NEOFLUAR ImmKorr NA 0.8, water) (all Zeiss) and a CoolSNAP HQ$^2$ high-resolution CCD camera (Photometrics). Bright-field images were acquired every second for 3 min (181 frames per movie). The movement of sporozoites was assessed using ImageJ 1.44o software (imagej.nih.gov/ij). The imaged sporozoites were clustered into three different categories (*Hegge et al., 2009*): 'continuously moving' sporozoites (consistently gliding with a speed of at least 0.2 µm/s over a minimum of 90 s); 'attached' sporozoites (displaying circular gliding motility, but failing to meet the 'moving' requirements or not showing any circular gliding but attached at either one or both ends); and 'floating' sporozoites (not attached to the glass surface). The speed of continuously moving sporozoites was determined using the manual tracking plug-in of ImageJ. For the analyses of the movement of salivary gland sporozoites, mosquitos were dissected on days 17–25 p.i. Motility was evaluated as described above with the following changes: images were taken every 3 s for 5 min (101 frames per movie) and productively moving sporozoites moved for at least 100 s. To measure the front-to-rear distance of moving sporozoites, a straight line was drawn connecting both ends using the straight line tool of ImageJ.

## In vitro sporozoite traversal, adhesion and invasion assays

WT (Fluo-frmg or cl15cy1), Δ*limp*-a, Δ*limp*-b and *limp::gfp* salivary gland sporozoites were dissected at day 21 p.i. To test traversal of human hepatoma cells in vitro, 5,000 salivary gland sporozoites of each parasite line were added, in the presence of 0.5 mg/mL of Dextran, Tetramethylrhodamine (Molecular Probes, #D1817), onto 110,000 Huh7 cells plated 16–20 hr before in 24-well plates and incubated for 2 hr at 37°C, 5% CO$_2$. Equivalent amounts of non-infected mosquito salivary gland debris were loaded as negative control. Cells were washed, trypsinised and analysed in a BD Biosciences FACSCalibur flow cytometer. Percentage of traversal was quantified as the ratio of Dextran-positive cells per live cells. To test the parasite capacity to adhere to and invade human hepatoma cells in vitro, 5,000 WT (Fluo-frmg or cl15cy1), Δ*limp*-a, Δ*limp*-b or *limp::gfp* salivary gland sporozoites were loaded onto 25,000 Huh7 cells plated 16–20 hr before in black-walled 96-well plates and incubated for 2 hr at 37°C, 5% CO$_2$. In the case of *limp::gfp*, sporozoites were also pre-incubated for 1 hr on ice in the presence of rabbit polyclonal anti-GFP (Abcam, #ab6556; 10 µg/mL) (RRID:AB_305564) or control rabbit IgGs (Abcam, #ab37415; 10 µg/mL) (RRID:AB_2631996) to evaluate possible invasion blockade. Cells were then fixed with 4% PFA/PBS and a dual colour anti-PbCSP immunofluorescence assay was performed to distinguish between sporozoites that invaded from those that did not, following published methods (*Rénia et al., 1988*). Briefly, external sporozoites were detected before cell permeabilisation using a 10 µg/mL suspension of 3D11 mouse anti-PbCSP (RRID:AB_2650479) (*Yoshida et al., 1980*) followed by goat anti-mouse IgG-Cy3 (Jackson ImmunoResearch Laboratories, Inc., #115-166-003; 1:500) (RRID:AB_2338699). Cells were then permeabilised with 0.1% TritonX-100/PBS and total sporozoites detected by a second incubation with the same 3D11 mouse anti-PbCSP dilution followed by goat anti-mouse IgG-Alexa Fluor488 (Jackson ImmunoResearch Laboratories, Inc., #115-546-006; 1:500) (RRID:AB_2338860). Cells were imaged under a Zeiss Axiovert 200M fluorescence microscope. Adhesion was quantified as the ratio of imaged sporozoites per number of Huh7 cells in the same optical fields.

## In vitro exoerythrocytic form (EEF) development assays

WT (Fluo-frmg or cl15cy1), $\Delta limp$-a, $\Delta limp$-b and limp::gfp salivary gland sporozoites were dissected at day 21 p.i. and tested for their ability to transform into EEFs inside Huh7 cells in culture. Of each parasite line, 5,000 salivary gland sporozoites were loaded onto 10,000 Huh7 cells plated 16–20 hr before in black-walled 96-well plates and incubated for 45 hr at 37°C, 5% $CO_2$. Cells were fixed with 4% PFA/PBS and an immunofluorescence (IFA) assay performed to detect parasite HSP70 and UIS4 (see below for IFA details). The area of cl15cy1 and limp::gfp EEFs were calculated based on HSP70 staining using ImageJ 1.47n software (imagej.nih.gov/ij).

## Live imaging and immunofluorescence assays (IFAs) of blood stages, ookinetes and sporozoites

Live imaging of transgenic limp::gfp parasites was done by retrieving ookinetes from mosquito blood meals at 16 h.p.i. To detect LIMP::GFP expression by IFA in blood stages, as well as in midgut and salivary gland sporozoites, mouse RBCs infected with limp::gfp parasites or dissected midgut and salivary gland sporozoites were stained with either rabbit polyclonal anti-GFP (Abcam, #ab6556; 1:500) (RRID:AB_305564), rabbit monoclonal anti-GFP (Life Technologies, #G10362) (RRID:AB_2536526), or mouse monoclonal anti-GFP (Abcam, #ab1218; 1:250) (RRID:AB_298911) as primary antibody. To detect mCherry::LIMP and LIMP::MYC, rabbit polyclonal anti-mCherry (Abcam, #ab183628) (RRID:AB_2650480) and mouse monoclonal anti-c-myc (Roche, #11667149001) (RRID: AB_390912) primary antibodies were used, respectively. As secondary antibody, goat anti-rabbit IgG-Alexa Fluor488 (Jackson ImmunoResearch Laboratories, Inc., #111-545-003; 1:500) (RRID:AB_2338046) or goat anti-mouse IgG-Alexa Fluor488 (Jackson ImmunoResearch Laboratories, Inc., #115-546-006; 1:500) (RRID:AB_2338860) was used. To detect CSP or TRAP in sporozoites, 3D11 mouse anti-PbCSP (RRID:AB_2650479) (*Yoshida et al., 1980*) (1:5,000) or rabbit polyclonal anti-PbTRAP repeats antiserum (RRID:AB_2650481) (*Ejigiri et al., 2012*) (1:1,000) were used. IFAs to detect HSP70 and UIS4 in in vitro-developed EEFs were done using parasite-specific 2E6 mouse monoclonal anti-PbHSP70 (RRID:AB_2650482) (*Tsuji et al., 1994*) at 18.75 µg/mL and goat polyclonal anti-PbUIS4 (SICGEN, #AB0042; 2 µg/mL) (RRID:AB_2333158/2333159), respectively, followed by donkey anti-mouse IgG-DyLight549 (Jackson ImmunoResearch Laboratories, Inc., #745-506-150; 1:500) (RRID:AB_2650483) and donkey anti-goat IgG-Alexa Fluor488 (Jackson ImmunoResearch Laboratories, Inc., #705-546-147; 1:500) (RRID:AB_2340430). For all IFAs, samples were fixed with 4% PFA/PBS for 10–60 min at RT, permeabilised (or not) with 0.1–0.2% Triton X-100/PBS for 10–20 min and blocked for 20–60 min at RT with 1–3% BSA/PBS. All antibody incubations were done in blocking solution for 45–60 min at RT and Hoechst-33342/PBS (1–5 µg/mL) or DRAQ5 (5 µM, Biostatus) was used to stain nuclei. Images were taken with Leica DM5000B and Zeiss Axiovert 200M fluorescence microscopes, as well as with a Leica TCS SP5 confocal microscope, and processed using ImageJ 1.47n software (imagej.nih.gov/ij).

## Transmission electron microscopy (TEM) and immuno-gold EM of sporozoites

Infected mosquito midguts and salivary glands were dissected on days 20–27 p.i and fixed for 3 hr at 4°C in 0.1 M cacodylate buffer, pH 7.4, containing 2.5% (v/v) glutaraldehyde. Following 1 hr of post-fixation with 1% (w/v) osmium tetroxide and 30 min of staining in 1% (w/v) uranyl acetate, samples were dehydrated in ethanol gradient (70–95–100%), transferred to propylene oxide and embedded in EPON resin. Semi-thin sections (300–400 nm) were stained with toluidine blue for light microscope evaluation. Ultra-thin sections (70 nm) were collected in Formvar-coated copper slot grids (AGAR Scientific) and then stained with 2% uranyl acetate and lead citrate (Reynolds recipe). For immuno-gold EM, ultra-thin sections (70 nm) collected in Formvar-coated Nickel 150 mesh grids were blocked for 30 min at RT with 0.8% BSA, 0.1% cold water fish skin gelatine, and incubated overnight at 4°C with rabbit polyclonal anti-GFP (Abcam, #ab6556; 1:50) (RRID:AB_305564). After a new blocking step of 15 min at RT, sections were incubated for 1 hr at RT with Protein A conjugated to 15 nm gold particles (obtained from Cell Microscopy Center, University Medical Center Utrecht, The Netherlands; 1:50). Sections were finally stained with 2% uranyl acetate and lead citrate (Reynolds recipe). Antibody and Protein A incubations were done in blocking solution. All grids were examined in a Hitachi H-7650 transmission electron microscope at 100kV acceleration.

## Multiple sequence alignments

Protein sequences in *Figure 1B* were retrieved from PlasmoDB (plasmodb.org) (RRID:SCR_013331) or from eupathdb.org (RRID:SCR_004512) for *Figure 1—figure supplement 1*. ClustalW alignments were performed at the EMBnet server (embnet.vital-it.ch/software/ClustalW.html) and shaded according to protein similarity levels with BOXSHADE 3.21 (www.ch.embnet.org/software/BOX_form.html) (RRID:SCR_007165).

## Homology modelling of LIMP

LIMP without signal peptide ($I^{23}$-$G^{110}$) was submitted to the QUARK online server (http://zhanglab.ccmb.med.umich.edu/QUARK/), an *ab initio* protein folding algorithm (*Xu and Zhang, 2012*). The model with the minimal TM score was chosen for all subsequent modelling experiments. To generate a model for LIMP::GFP, chain A of the GFP structure (PDB ID: 1GFL) was chosen, linked to LIMP using the Prime Module of the Schrödinger package (www.schroedinger.com) (RRID:SCR_014879) and used to develop a three-dimensional structure of LIMP::GFP. Analogously, mCherry (PDB ID: 2H5Q) was chosen for modelling mCherry::LIMP.

## Molecular dynamics

Modelled LIMP, LIMP::GFP, mCherry::LIMP and LIMP::MYC were prepared for all-atom molecular dynamics (MD) simulations using the tleap program within the AMBER molecular dynamics package version 14 (*Case et al., 2014*) (http://ambermd.org) (RRID:SCR_014230); topology and parameters were generated using the ff99SBildn force field (*Lindorff-Larsen et al., 2010*); proteins were solvated using TIP3P water-model in truncated octahedron box type. Standard settings within PMEMD (Particle mesh Ewald molecular dynamics) were then used for the simulations following a two-step energy minimisation: minimisation while keeping restrain on the protein atoms (1,500 steps of steepest descent method), and all-atom minimisation (first 2,500 steps of steepest descent, next 1,000 steps of conjugate gradient). Minimised systems were then gradually heated (0 to 298 Kelvin in 100 picoseconds) using canonical ensemble (NVT). In the next step, we equilibrated pre-heated system in isothermal–isobaric ensemble (NPT) at 298 Kelvin: Berendsen temperature coupling and a constant pressure of 1 atm with isotropic molecule-based scaling was used in the equilibration. To reduce simulation run time, the SHAKE algorithm was applied to fix all covalent bonds containing hydrogen atoms. A 10 Å cut-off was applied to treat nonbonding interactions such as short-range electrostatics and van-der-Waals interactions, while the particle-mesh-Ewald (PME) method was used for long-range electrostatic interactions. Simulations were carried out with periodic boundary condition in NPT ensemble for 100 ns. The analyses of MD simulations were carried out by CPPTRAJ module of AMBER 14. Visual Molecular Dynamics (VMD) (RRID:SCR_001820) (*Humphrey et al., 1996*) and PyMOL (www.pymol.org) (RRID:SCR_000305) was used for visualisation.

## Statistical methods

Statistical analyses were performed using Prism software package 5 or higher (GraphPad Software) (RRID:SCR_002798).

## Acknowledgements

We would like to thank Ana Guerreiro, Ana Parreira and Fernanda Baptista (iMM Lisboa), as well as Jai Ramesar and Johannes Kroeze (LMRG, LUMC) for management and experimental support; Leonor Pinho and Miriam Reinig for mosquito production; Drs. Céline Carret and António Mendes (iMM Lisboa) for technical advice; Dr. Photini Sinnis (JHMRI, JHSPH, USA) for providing the anti-PbTRAP antiserum; Dr. Erin Tranfield (IGC EM Facility, Portugal) for technical and intellectual input during EM experiments; and access to bwGRiD (http://www.bw-grid.de), member of the German D-Grid initiative, funded by the Ministry for Education and Research (Bundesministerium für Bildung und Forschung) and the Ministry for Science, Research and Arts Baden-Wuerttemberg (Ministerium für Wissenschaft, Forschung und Kunst Baden-Württemberg).

# Additional information

## Funding

| Funder | Grant reference number | Author |
|---|---|---|
| Fundação para a Ciência e a Tecnologia | PTDC/BIA-BCM/105610/2008 | Gunnar R Mair |
| Fundação para a Ciência e a Tecnologia | PTDC/SAU-MIC/122082/2010 | Gunnar R Mair |
| Fundação para a Ciência e a Tecnologia | SFRH/BD/63849/2009 | Jorge M Santos |
| Fundação para a Ciência e a Tecnologia | BPD/81953/2011 | Vanessa Zuzarte-Luís |
| Horizon 2020 Framework Programme | Marie Sklodowska-Curie grant agreement No 660211 | Saskia Egarter |
| Horizon 2020 Framework Programme | ERC-SG 281719 | Friedrich Frischknecht |
| Human Frontier Science Program | RGY0071½011 | Friedrich Frischknecht |
| Chica and Heinz Schaller Foundation | | Friedrich Frischknecht |
| Seventh Framework Programme | EviMalar | Friedrich Frischknecht |

The funders had no role in study design, data collection and interpretation, or the decision to submit the work for publication.

## Author contributions

JMS, Conceptualization, Data curation, Formal analysis, Supervision, Funding acquisition, Investigation, Methodology, Writing—original draft, Project administration, Writing—review and editing, Conception and design, Acquisition of data, Analysis and interpretation of data, Drafting and revising the article, Final approval of the version to be published; SE, Conceptualization, Data curation, Formal analysis, Funding acquisition, Investigation, Methodology, Writing—original draft, Writing— review and editing, Conception and design, Acquisition of data, Analysis and interpretation of data, Drafting and revising the article, Final approval of the version to be published; VZ-L, Conceptualization, Formal analysis, Funding acquisition, Investigation, Methodology, Writing—review and editing, Conception and design, Acquisition of data, Analysis and interpretation of data, Drafting and revising the article, Final approval of the version to be published; HK, Conceptualization, Formal analysis, Funding acquisition, Investigation, Methodology, Acquisition of data, Analysis and interpretation of data; CAM, JK, AP, Conceptualization, Formal analysis, Investigation, Methodology, Acquisition of data, Analysis and interpretation of data; MdC, BF-F, Investigation, Acquisition of data, Analysis and interpretation of data; CJJ, FF, Investigation, Methodology, Acquisition of data, Analysis and interpretation of data; GRM, Conceptualization, Data Curation, Formal analysis, Supervision, Funding acquisition, Investigation, Methodology, Writing—original draft, Project administration, Writing— review and editing, Conception and design, Acquisition of data, Analysis and interpretation of data, Drafting and revising the article, Final approval of the version to be published

## Author ORCIDs

Jorge M Santos, http://orcid.org/0000-0002-4940-5207
Jessica Kehrer, http://orcid.org/0000-0001-5084-3485
Blandine Franke-Fayard, http://orcid.org/0000-0001-6041-4182
Gunnar R Mair, http://orcid.org/0000-0002-5187-3691

## Ethics

Animal experimentation: In Instituto de Medicina Molecular (iMM, Lisbon, Portugal), animal experimentation protocols were approved by the iMM Animal Ethics Committee (under authorisation AEC_2010_018_GM_Rdt_General_IMM) and the Portuguese authorities (Direção Geral de

Alimentação e Veterinária). Animal experiments performed in Leiden University Medical Center (LUMC, Leiden, The Netherlands) were approved by the Animal Experiments Committee of the Leiden University Medical Center (DEC 10099; 12042; 12120). At the University of Heidelberg Medical School (Heidelberg, Germany) animal work was approved by the German authorities (Regierungspräsidium Karlsruhe; 35-9185.81/G-266/16) and performed in compliance with FELASA guidelines and regulations. All animal work was in accordance with EU regulation. Animals used were female BALB/c, OF-1 and NMRI mice (6-8 weeks old) bred at Charles River or Janvier, France.

## Additional files

### Supplementary files

• Supplementary file 1. Summary of Plasmodium surface-, microneme- or rhoptry-localised proteins with function in ookinete and sporozoite motility, adhesion, cell traversal and cell invasion.

• Supplementary file 2. Primers used in the present work. (A) Primers used in life-cycle RT-PCRs and qPCR of RNA extracts from infected livers. (B) Primers used in the generation of gene deletion, GFP-tagging, mCherry-tagging and MYC-tagging constructs. (C) Primers used in genotyping and RT-PCR of mutant parasite lines.

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
