## [Decision Letter]

[Editors’ note: a previous version of this study was rejected after peer review, but the authors submitted for reconsideration. The first decision letter after peer review is shown below.]

Thank you for choosing to send your work entitled "*Plasmodium* LIMP protein regulates adhesion site turnover, gliding motility and infection of mosquito and mammalian hosts" for consideration at *eLife*. Your article has been favorably evaluated by Prabhat Jha (Senior Editor) and three reviewers, one of whom, Urszula Krzych, is a member of our Board of Reviewing Editors. Based on our discussions and the individual reviews below, we regret to inform you that your work will not be considered for publication in *eLife*. However,we strongly encourage you to return a new submission upon completion of the proposed experiments.

Summary:

The manuscript by Santos et al. describes interesting findings on LIMP involvement in sporozoite adhesion and motor driven gliding motility, a unique feature found in malaria parasite. Specifically, the authors identified LIMP as a key regulator of sporozoite adhesion during gliding motility in *Plasmodium berghei*. The aim was to demonstrate that during sporogony, LIMP localizes to the sporozoite surface where it regulates parasite attachment to and detachment from the substrate. Transcripts of LIMP are abundantly detected in gametocytes associated with the translational repressors DOZI and CITH, and in young, but not old, ookinetes, in oocysts and in sporozoites. However, translation of LIMP, as inferred by expression of a LIMP:GFP fusion, is restricted to ookinetes, oocysts and sporozoites, indicating that LIMP expression is translational repressed in gametocytes and LIMP mRNA is maternally inherited in ookinetes.

The role of LIMP was first assessed by targeted gene disruption. In the mosquito, a decrease in the number of LIMP- oocysts was observed in the midgut, and of LIMP- sporozoites inside the salivary glands, but not inside the oocysts or in the hemolymph when normalized by the number of oocysts, indicating a role of LIMP in the crossing of epithelial barriers. In mice, LIMP- sporozoites are incapable of establishing liver and blood infection. In vitro, the mutant sporozoite presents an impaired motility, and consequently, a defect in host cell traversal and invasion.

The paper contains a large arsenal of sophisticated molecular approaches and well conducted parasitological and cell biological techniques. However, the major conclusion, namely that expression of LIMP on the parasite surface is required for gliding motility, cell traversal, regulation of adhesion site turnover, and hepatocyte invasion, is not convincingly supported by the data presented in this manuscript. The interpretation of the results of the role of LIMP in the regulation of adhesion site turnover in *Plasmodium*, the flaws in the experimental design, the lack of essential tools, and the choice of inappropriate techniques diminish the value of the study.

Essential revisions:

1) There isn't any evidence for native LIMP being present on the parasite surface. Considering that GFP (27 kDA) has more than double the molecular weight of LIMP (13 kDa), there is a concern that the resulting fusion protein is normally processed and intracellularly trafficked like the native LIMP. Data supporting this conclusion came from experiments using LIMP:GFP sporozoites, where is not possible to determine if the observed LIMP phenotype is due to a specific modification of putative binding sites of LIMP as suggested by the authors, or by a non-specific steric hindrance of GFP.

A) An elegant way to address this point would be to generate point mutations in LIMP to mimicry the destabilization caused by the GFP fusion, discarding a non-specific effect of GFP.

B) Alternatively, the dynamic gliding analysis of the population of LIMP- sporozoites that were able to glide and left behind 1 to 10 trails (30% of deltaA population, Figure 6) could bring direct data about the loss-of-function of LIMP.

C) Another possibility and probably the most interesting would be to analyze the gliding pattern of LIMP- ookinetes, since gene deletion decreased in > 2 fold the number of oocysts in the midgut. The authors showed that thought this gene is transcribed in gametocyte but is translationally repressed and only expressed in ookinete. However, they show that the ookinete conversion is normal but the sporozoite function is affected. It is important that authors show that whether the gliding motility of ookinete is affected at all. Please provide information with respect to the translationally repressed protein involved in ookinete invasion and where does LIMP fits in that hierarchy.

D) Also, tiny fluorescent tags have been developed that could have been used for visualization of LIMP. In my opinion, biological consequences caused by GFP major alteration of LIMP cannot be excluded. The fact that addition of GFP to the C-terminus of LIMP led to a loss of structural integrity in the fusion protein supports this view. The sentence "fusion of GFP to LIMP seems to slightly interfere with normal function" (Discussion, third paragraph) may be an understatement.

E) Although the authors generated LIMP-GFP parasites, results from western blot with GFP antibody are needed to show if the protein is rightly expressed. This is important for any tagged line generated.

2) The immunoelectron microscopy technique, which aims to demonstrate the presence of LIMP on the sporozoite surface, is completely inadequate. Using Epon sections of tissue that was fixed with 2.5% glutaraldehyde, postfixed with 1% osmium tetroxide, stained with 1% uranyl acetate, immersed in solvents, and subjected high temperature embedding has little to no chance of yielding any physiologically relevant signal. The authors claim that established immunoelectron microscopy approaches fail to show membranes (Discussion, second paragraph). This is not correct. In fact, ultracryosections are known to preserve and show membranes extremely well, in particular when combined with negative staining of the immunolabeled section or state-of-the-art positive staining techniques. Further, what was the reason for choosing 15 nm protein A gold particles (not immuno-gold)? The larger the particle, the smaller the signal. It would have been much more reasonable to use 5 nm immunogold or, even better, 1 nm immunogold plus silver enhancement. Perhaps this would have allowed the authors to present more than the few gold particles shown in Figure 8.

3) Throughout the study, there is no direct immunofluorescence labeling of LIMP of the parasite surface. LIMP-specific antibodies did not work. The detection of LIMP relies on GFP-specific antibodies. For these reasons, the claim "In sporozoites, LIMP is present at the parasite PM, as determined by immuno-gold studies" (Discussion, second paragraph) is not justified.

4) It should be emphasized that the detailed description of sporozoite adhesion and motility (Discussion, third paragraph) is solely based on in vitro interactions with artificial substrates. One of the binding partners in such interactions, in this study antibodies, is irreversibly immobilized on a completely flat surface, which allows visualization of a trail. Of note, CSP-containing trails have not been observed in sporozoite-exposed cell cultures. The situation in live three-dimensional tissues with their fluid cell membranes and a multitude of extracellular matrix components is even more different. As a consequence, sporozoites are highly flexible and move quite differently in natural environments such as mosquito salivary glands and ducts or solid tissues such as liver.

5) If the sporozoite motility is affected and they are not able to invade the salivary gland then the authors should show the processing of CSP protein, as it is the most important protein expressed on the surface of sporozoites that is involved in sporozoite invasion and motility. Whether or not LIMP affects CSP processing is important to dissect out. It is crucial to know as we have seen in number of mutants like ECP etc. that CSP processing is affected. In addition to this, please address the issue of pattern of UIS 4 expression in these sporozoites.

[Editors’ note: what now follows is the decision letter after the authors submitted for further consideration.]

Thank you for resubmitting your work entitled "Malaria parasite LIMP regulates adhesion site turnover during gliding motility and infection of mosquito and mammal host" for further consideration at *eLife*. Your revised article has been favorably evaluated by Prabhat Jha as the Senior Editor, Urszula Krzych as the Reviewing Editor, and three reviewers.

The manuscript has been improved but there are some remaining issues that need to be addressed before acceptance, as outlined below:

This manuscript describes for the first time the function of *Plasmodium berghei* LIMP protein, which appears to have a role in both the mosquito and the mammalian host during sporozoite motility. According to the authors hypothesis, LIMP is crucial for sporozoite invasion of salivary glands and hepatocytes via orchestration of adherence. The reviewers concluded that the work describes new and important findings and that the technically sound experiments were conducted with great care. In addition, they concluded that the previous reviewer's concerns have been adequately addressed.

The concerns that arose could be corrected by revising and editing the current version of the manuscript. The main concern expressed by all reviewers was the issue with overstating the findings regarding the essentiality for LIPM for the initial attachment of sporozoites to the substrate and LIMPs role in regulating efficient turnover of adhesion sites. The current interpretation should be toned down and the title modified accordingly. Additionally, some results could be omitted as they appear unnecessary in support of the authors conclusions. For example, the ookinetes data is unnecessary as regards the LIMP localization and motility phenotypes; the dermal inoculation data is also unconvincing and can be omitted as intravenous inoculation shows a drastic phenotype on its own.

*Reviewer #1:*

This manuscript describes new and important findings. The work is technically sound and thorough, and experiments conducted with great care. The previous reviewer's concerns have been adequately addressed. I recommend publication in *eLife*.

*Reviewer #2:*

This paper nicely characterizes the LIMP protein of *Plasmodium berghei*, a protein that until now had no known function. The authors describe LIMPs role in two hosts: mosquito and mammalian host, predominately through the utilization of a knockout parasite line, as well as describe the phenotypes of protein-tagged lines and perform modeling. Through these means, the authors demonstrate a role for LIMP in sporozoite motility. However, its role in ookinete and oocyst development are less clear. The authors hypothesize that LIMP is crucial for sporozoite invasion (of salivary glands and hepatocytes) through the orchestration of adherence. The authors finally suggest that the helices of LIMP are involved in interactions with this protein and/or protein confirmation.

Concern arises in overstatement of findings and the lack of depth in describing the role of LIMP in motility. Additionally, a few experimental concerns are presented, but in most cases this reviewer feels the data could just be omitted as the phenotypes are clear or unnecesary. For example, this reviewer believes the ookinetes data is unnecessary and creates confusion towards the LIMP localization and motility phenotypes. Additionally, the dermal inoculation data is unconvincing and can be omitted as intravenous inoculation shows a drastic phenotype on its own.

Of largest concern is that the overall conclusion of this paper that "LIMP is essential for the initial attachment of sporozoites to the substrate, and regulates efficient turnover of attachment sites in gliding parasites" (Discussion, second paragraph) is not supported. This reviewer believes this is an overstatement of the findings and believes more in depth analysis of motility, such as reflection interference contrast microscopy, is required to know what is happening with the adhesion sites. The conclusions herein are an oversimplification of possible LIMP functions during sporozoite motility and invasion.

*Reviewer #3:*

The manuscript entitled “Malaria parasite LIMP regulates adhesion site turnover during gliding motility and infection of mosquito and mammal host” by Mair et al., presents in the protozoan *Plasmodium berghei* a detailed phenotype characterization when the newly identified LIMP protein is missing. Using the engineered LIMP mutants, this characterization leads to uncover the regulatory function of LIM in the motile and invasive, but not developmental properties of *Plasmodium* (i) ookinetes, the stage that generates oocysts in the haemolymph of the hosting mosquitoes and (ii) sporozoites, the stage formed within the oocyts and that next reach the salivary glands from where they can be injected together with salivia to the mammalian hosts through mosquito blood feeding. LIMP appears also as a major determinant for sporozoite invasiveness of mammalian liver cells in vivo and in vitro.

To support these conclusions, Mair et al., have used a combination of relevant molecular genetics to engineer Δlimp and LIMP::GFP, mcherry::limp and limp::myc (at the endogenous locus) strains in these haploid parasites, in conjunction with several microscopy assays (from live imaging to EM) to convincingly provide solid data of the regulatory functions of LIMP. The text is generally clear and the experimental designs sound appropriate, with precise conclusions and no overstatements.

I understood that my review comes after the first round of the reviewing process of this manuscript. I have not seen the initial version but I feel this one to be of enough quality for recommending its publication in *eLife*.

---

## [Author Response]

[Editors’ note: the author responses to the first round of peer review follow.]

*Essential revisions:*

*1) There isn't any evidence for native LIMP being present on the parasite surface. Considering that GFP (27 kDA) has more than double the molecular weight of LIMP (13 kDa), there is a concern that the resulting fusion protein is normally processed and intracellularly trafficked like the native LIMP. Data supporting this conclusion came from experiments using LIMP:GFP sporozoites, where is not possible to determine if the observed LIMP phenotype is due to a specific modification of putative binding sites of LIMP as suggested by the authors, or by a non-specific steric hindrance of GFP.*

We always argued that the limping phenotype was a serendipitous effect caused by the GFP tag; while limp KO parasites fail to attach to the surface preventing any further analysis, the GFP tag produced and exposed a detachment phenotype and function for LIMP. LIMP is thus – as we detail – required for smooth adhesion site turnover allowing uninterrupted gliding. A key finding is that the *limp::gfp* parasite (here LIMP::GFP is the sole source for LIMP as we tagged the endogenous gene in this haploid protozoan) produces infectious sporozoites while the KO does not.

In the revised manuscript we have now explored the matter of protein localization and resulting phenotypes in several new ways:

1) We provide the missing Western blot for LIMP::GFP showing that the fusion protein is produced as expected.

2) To address the localization of LIMP we generated 2 additional transgenic lines: LIMP tagged at the C-terminus with a much smaller MYC tag; and LIMP tagged at the N-terminus with mCherry. Both lines replicate the staining pattern found for LIMP::GFP (a comparison of the 3 lines is shown in Figure 4 and SOM Figure 6). We are thus confident to have localised the protein correctly; all 3 protein tags show the same localization and each of the mutants is infectious to mosquito in mouse while the KO is not.

3) We analysed the gliding motility behaviour of each line: the data show that the limping defect is specific for LIMP tagged at the C-terminus with GFP (see Figure 5).

4) We next modelled all of the mutants and wildtype: the results show that both LIMP helices are occluded by the C- terminal GFP tag; MYC and mCherry tags do not affect the accessibility of this surface (see Figure 6, SOM Figures 9 and

10) suggesting that an interaction between parasite and substrate is required for smooth, uninterrupted gliding.

In summary: we provide 3 transgenic lines to address the localization of LIMP producing consistent, surface localisation in a manner akin to TRAP. It is also important to stress that LIMP::GFP parasites, although showing a slight motility defect that allowed us to expand on LIMP’s function, complete their life cycle just as well as wildtype parasites (see SOM Figure 3 and Figure 6). If the GFP tag were to prevent correct processing and trafficking of LIMP, one would expect to see an impairment in life cycle progression, mimicking the LIMP knock-out phenotype. Therefore, we are confident that LIMP::GFP is behaving as native, wildtype LIMP. In all 3 mutant lines, the tagged LIMP protein is the sole source for LIMP as we routinely tag the endogenous gene (not expressing fusion proteins from plasmid constructs). We have attempted the production of peptide antibodies against native LIMP but failed to produce sera that provide specificity and reproducibility, preventing us from confirming subcellular localization of native, untagged LIMP. We are however confident that localization by tagging the endogenous locus in 3 different ways and 3 identical outcomes provides clear evidence for LIMP localization to the surface of the parasite.

*A) An elegant way to address this point would be to generate point mutations in LIMP to mimicry the destabilization caused by the GFP fusion, discarding a non-specific effect of GFP.*

The generation of point mutations has not been possible so far. We experience reversion to wildtype during plasmid construct integration into the endogenous locus.

*B) Alternatively, the dynamic gliding analysis of the population of LIMP- sporozoites that were able to glide and left behind 1 to 10 trails (30% of deltaA population, Figure 6) could bring direct data about the loss-of-function of LIMP.*

Figure 4 of the new submission shows a dynamic analysis of gliding motility capability of LIMP KO sporozoites; with the vast majority of parasites not attaching at all nor exhibiting continuous movement the loss-of-function phenotype is “failure to attach”.

*C) Another possibility and probably the most interesting would be to analyze the gliding pattern of LIMP- ookinetes, since gene deletion decreased in > 2 fold the number of oocysts in the midgut. The authors showed that thought this gene is transcribed in gametocyte but is translationally repressed and only expressed in ookinete. However, they show that the ookinete conversion is normal but the sporozoite function is affected. It is important that authors show that whether the gliding motility of ookinete is affected at all. Please provide information with respect to the translationally repressed protein involved in ookinete invasion and where does LIMP fits in that hierarchy.*

We have now quantified gliding motility in ookinetes of the LIMP KO mutant and find no difference to wildtype in speed (SOM Figure 5). It thus seems unlikely that the oocyst reduction phenotype seen for LIMP KO parasites is due to impaired motility. The ookinete may require LIMP for optimal midgut traversal. However, the main phenotypes (gliding motility, infectiousness) were observed at the sporozoite stage and for that reason we focused our work on characterising the role of LIMP in this life cycle stage.

As to the translation of LIMP in the ookinete: our data show that LIMP—evident from the limp KO—is not required for ookinete formation. While a global interference with translational repression precludes the formation of ookinetes altogether, very few individual genes have been addressed in the context of ookinete morphogenesis or function. We have recently published on the role of 2 DHHC S-acyl-transferases, defined the novel CPW-WPC gene family, and explored protein function of translationally repressed gene products in a 2014 Genome Biology paper. Our data indicate that the function of maternally provided transcripts covers a vast developmental window of the malaria parasite: starting from early zygote transformation to the establishment of ookinete-specific crystalloid organelle and its function during sporozoite formation in the oocyst.

*D) Also, tiny fluorescent tags have been developed that could have been used for visualization of LIMP. In my opinion, biological consequences caused by GFP major alteration of LIMP cannot be excluded. The fact that addition of GFP to the C-terminus of LIMP led to a loss of structural integrity in the fusion protein supports this view. The sentence "fusion of GFP to LIMP seems to slightly interfere with normal function" (Discussion, third paragraph) may be an understatement.*

We have produced 2 more lines with tagged LIMP: (1)mCherry::LIMP and (2) LIMP::MYC. The localization of these fusion proteins mimicks the one found for LIMP::GFP. We have removed the sentence on ‘normal function’ as it is misleading when GFP-tagging led to the discovery of the ‘limping’ phenotype.

*E) Although the authors generated LIMP-GFP parasites, results from western blot with GFP antibody are needed to show if the protein is rightly expressed. This is important for any tagged line generated.*

We have added Western blot data showing that the tagged protein is of the correct size (see Figure 1).

*2) The immunoelectron microscopy technique, which aims to demonstrate the presence of LIMP on the sporozoite surface, is completely inadequate. Using Epon sections of tissue that was fixed with 2.5% glutaraldehyde, postfixed with 1% osmium tetroxide, stained with 1% uranyl acetate, immersed in solvents, and subjected high temperature embedding has little to no chance of yielding any physiologically relevant signal. The authors claim that established immunoelectron microscopy approaches fail to show membranes (Discussion, second paragraph). This is not correct. In fact, ultracryosections are known to preserve and show membranes extremely well, in particular when combined with negative staining of the immunolabeled section or state-of-the-art positive staining techniques. Further, what was the reason for choosing 15 nm protein A gold particles (not immuno-gold)? The larger the particle, the smaller the signal. It would have been much more reasonable to use 5 nm immunogold or, even better, 1 nm immunogold plus silver enhancement. Perhaps this would have allowed the authors to present more than the few gold particles shown in Figure 8.*

We understand that by ultracryosectioning our samples we could have obtained good preservation of membranes without subjecting them to harsh processing conditions. However, a cryostat was not available to us at the time this work was developed. After discussing with highly renowned experts in electron microscopy, particularly in immuno-gold electron microscopy, namely Dr. Erin Tranfield from Instituto Gulbenkian de Ciência, we decided to conduct our experiments in order to maximise membrane preservation with the techniques that were available to us. This was unfortunately at the expense of possibly losing some of the epitopes of interest or rendering them inaccessible to antibody binding. Nonetheless, it was still possible to draw a clear conclusion about the subcellular localisation of LIMP. Having in mind the limitations of the methodology employed, we very carefully conducted three independent experiments, including detailed control conditions alongside with our actual experimental settings. For instance, we included controls where non GFP-expressing parasites were subjected to the exact same staining protocol and also LIMP::GFP sporozoite samples where the primary antibody step was omitted. Average gold particle distribution was as follows: LIMP::GFP parasites subjected to the full staining protocol – 76% at the plasma membrane, 11% extracellular, 13% intracellular –; LIMP::GFP parasites omitting primary anti-GFP antibody – 26% at plasma membrane, 42% extracellular, 32% intracellular –; WT parasites subjected to full staining protocol – 38% at plasma membrane, 50% extracellular, 12% intracellular. It is also important to note that the absolute number of gold particles found on each type of sample was also very indicative of specific staining, as a total of 281 particles were counted in LIMP::GFP parasites subjected to the full staining protocol, while only a mere 19 and 8 particles were found in LIMP::GFP parasites omitting primary anti-GFP antibody and WT parasites subjected to full staining protocol, respectively. Finally, we tested different sizes of gold particles and secondary antibodies. We observed for instance that 6 nm gold particles were too small in relation to the size of the sporozoite’s subcellular structures. 6 nm is just about the size of the plasma membrane cross section and very easily confounded with other electron dense structures in our samples. We have thus determined that 15 nm is the ideal size for a gold particle to be used in this particular type of sample. Also, protein A is widely used in immuno-electron microscopy for its known high specificity and avidity to certain types of immunoglobulins. Protein A binds very strongly to human, mouse and rabbit IgG, but is almost unreactive to rat IgG. In this study we used as primary antibody a rabbit IgG. In our hands, using protein A did not decrease specificity when compared to conventional secondary anti-rabbit antibodies and was thus selected for further experiments.

*3) Throughout the study, there is no direct immunofluorescence labeling of LIMP of the parasite surface. LIMP-specific antibodies did not work. The detection of LIMP relies on GFP-specific antibodies. For these reasons, the claim "In sporozoites, LIMP is present at the parasite PM, as determined by immuno-gold studies" (Discussion, second paragraph) is not justified.*

We are confident to have addressed localization reliably through the generation of additional mutants. LIMP::GFP, LIMP::MYC and mCherry::LIMP produce the same staining patterns and support infection of the mosquito and rodent hosts, while the KO does not.

In summary: we provide 3 transgenic lines to address the localization of LIMP producing consistent, surface localisation in a manner akin to TRAP.

*4) It should be emphasized that the detailed description of sporozoite adhesion and motility (Discussion, third paragraph) is solely based on in vitro interactions with artificial substrates. One of the binding partners in such interactions, in this study antibodies, is irreversibly immobilized on a completely flat surface, which allows visualization of a trail. Of note, CSP-containing trails have not been observed in sporozoite-exposed cell cultures. The situation in live three-dimensional tissues with their fluid cell membranes and a multitude of extracellular matrix components is even more different. As a consequence, sporozoites are highly flexible and move quite differently in natural environments such as mosquito salivary glands and ducts or solid tissues such as liver.*

We are aware that sporozoite motility in the context of a three-dimensional tissue can be different from that seen on bi- dimensional surfaces such as glass slides or cell monolayers. Nonetheless, the techniques employed in this study are well-established, standardized and thoroughly validated in vitro models used routinely to analyse and compare motility, traversal and invasion phenotypes of malaria sporozoites. We compare – always directly – the different mutants with wildtype lines; the results we are obtaining are thus highly relevant.

We have sometimes seen CSP-trails in human hepatoma monolayer cultures that were not coated with anti-CSP antibodies previously to sporozoite seeding. This would indicate that sporozoites do shed CSP on the surface on which they glide, irrespectively of whether that surface was pre-treated to capture/trap CSP molecules. This suggests that the analysis of CSP trail-based sporozoite motility might be more physiologically relevant than anticipated.

*5) If the sporozoite motility is affected and they are not able to invade the salivary gland then the authors should show the processing of CSP protein, as it is the most important protein expressed on the surface of sporozoites that is involved in sporozoite invasion and motility. Whether or not LIMP affects CSP processing is important to dissect out. It is crucial to know as we have seen in number of mutants like ECP etc. that CSP processing is affected. In addition to this, please address the issue of pattern of UIS 4 expression in these sporozoites.*

CSP processing is an interesting point, but given the lack of indication that there is a defect—KO parasites do produce and traffic CSP to the parasite surface, and, albeit rarely, deposit trails onto the microscope slide surface—there is no indication that processing should be affected. This is shown in Figure 3.

We do not understand the reference to the UIS4 expression issue.

[Editors' note: the author responses to the re-review follow.]

*[…] The concerns that arose could be corrected by revising and editing the current version of the manuscript. The main concern expressed by all reviewers was the issue with overstating the findings regarding the essentiality for LIPM for the initial attachment of sporozoites to the substrate and LIMPs role in regulating efficient turnover of adhesion sites. The current interpretation should be toned down and the title modified accordingly. Additionally, some results could be omitted as they appear unnecessary in support of the authors conclusions. For example, the ookinetes data is unnecessary as regards the LIMP localization and motility phenotypes; the dermal inoculation data is also unconvincing and can be omitted as intravenous inoculation shows a drastic phenotype on its own.*

As suggested, we have toned down our conclusions regarding the role of LIMP in initial attachment of sporozoites and regulation of adhesion site turnover throughout the manuscript and also the title. Most of the ookinete data, particularly LIMP knock-out gliding motility was removed, together with transmission data through mosquito bite.

*Reviewer #1:*

*This manuscript describes new and important findings. The work is technically sound and thorough, and experiments conducted with great care. The previous reviewer's concerns have been adequately addressed. I recommend publication in eLife.*

We are very thankful for the positive feedback and appreciation of our work.

*Reviewer #2:*

*This paper nicely characterizes the LIMP protein of Plasmodium berghei, a protein that until now had no known function. The authors describe LIMPs role in two hosts: mosquito and mammalian host, predominately through the utilization of a knockout parasite line, as well as describe the phenotypes of protein-tagged lines and perform modeling. Through these means, the authors demonstrate a role for LIMP in sporozoite motility. However, its role in ookinete and oocyst development are less clear. The authors hypothesize that LIMP is crucial for sporozoite invasion (of salivary glands and hepatocytes) through the orchestration of adherence. The authors finally suggest that the helices of LIMP are involved in interactions with this protein and/or protein confirmation.*

*Concern arises in overstatement of findings and the lack of depth in describing the role of LIMP in motility. Additionally, a few experimental concerns are presented, but in most cases this reviewer feels the data could just be omitted as the phenotypes are clear or unnecesary. For example, this reviewer believes the ookinetes data is unnecessary and creates confusion towards the LIMP localization and motility phenotypes. Additionally, the dermal inoculation data is unconvincing and can be omitted as intravenous inoculation shows a drastic phenotype on its own.*

Conclusions inferred from experimental results were toned down throughout the manuscript. Most of the ookinete data, particularly LIMP knock-out gliding motility was removed, together with transmission data through mosquito bite.

*Of largest concern is that the overall conclusion of this paper that "LIMP is essential for the initial attachment of sporozoites to the substrate, and regulates efficient turnover of attachment sites in gliding parasites" (Discussion, second paragraph) is not supported. This reviewer believes this is an overstatement of the findings and believes more in depth analysis of motility, such as reflection interference contrast microscopy, is required to know what is happening with the adhesion sites. The conclusions herein are an oversimplification of possible LIMP functions during sporozoite motility and invasion.*

Overstatements such as the one pointed out above have been rephrased and toned down. Namely, the sentence in the second paragraph of the Discussion and the title of the manuscript have been modified in line with this reviewer’s comments.

*Reviewer #3:*

*[…] I understood that my review comes after the first round of the reviewing process of this manuscript. I have not seen the initial version but I feel this one to be of enough quality for recommending its publication in eLife.*

We very much appreciate the reviewer’s comments and are thankful for the positive feedback.